EMBO
Molecular Medicine

# Targeting the pregnane X receptor using microbial metabolite mimicry

Zdeněk Dvořák[1,†,*] (iD), Felix Kopp[2,†], Cait M Costello[3], Jazmin S Kemp[3], Hao Li[4,†], Aneta Vrzalová[1,†], Martina Štěpánková[1], Iveta Bartoňková[1], Eva Jiskrová[1], Karolína Poulíková[1], Barbora Vyhlídalová[1], Lars U Nordstroem[2], Chamini V Karunaratne[2], Harmit S Ranhotra[4,‡], Kyu Shik Mun[5], Anjaparavanda P Naren[5], Iain A Murray[6], Gary H Perdew[6], Julius Brtko[7], Lucia Toporova[7], Arne Schön[8], Bret D Wallace[9,§], William G Walton[9], Matthew R Redinbo[9], Katherine Sun[10], Amanda Beck[11], Sandhya Kortagere[12,**] (iD), Michelle C Neary[13], Aneesh Chandran[14], Saraswathi Vishveshwara[14], Maria M Cavalluzzi[15], Giovanni Lentini[15], Julia Yue Cui[16], Haiwei Gu[17], John C March[3], Shirshendu Chatterjee[18], Adam Matson[19], Dennis Wright[20], Kyle L Flannigan[21], Simon A Hirota[21], Ryan Balfour Sartor[22] & Sridhar Mani[4,***] (iD)

## Abstract

The human PXR (pregnane X receptor), a master regulator of drug metabolism, has essential roles in intestinal homeostasis and abrogating inflammation. Existing PXR ligands have substantial off-target toxicity. Based on prior work that established microbial (indole) metabolites as PXR ligands, we proposed microbial metabolite mimicry as a novel strategy for drug discovery that allows exploiting previously unexplored parts of chemical space. Here, we report functionalized indole derivatives as first-in-class non-cytotoxic PXR agonists as a proof of concept for microbial metabolite mimicry. The lead compound, FKK6 (Felix Kopp Kortagere 6), binds directly to PXR protein in solution, induces PXR-specific target gene expression in cells, human organoids, and mice. FKK6 significantly represses pro-inflammatory cytokine production cells and abrogates inflammation in mice expressing the human PXR gene. The development of FKK6 demonstrates for the first time that microbial metabolite mimicry is a viable strategy for drug discovery and opens the door to underexploited regions of chemical space.

**Keywords** drugs; microbial metabolite; mimics; pregnane X receptor; therapy

1 Department of Cell Biology and Genetics, Palacký University, Olomouc, Czech Republic
2 Department of Biochemistry, Albert Einstein College of Medicine, Bronx, NY, USA
3 The Department of Biological and Environmental Engineering, Cornell University, Ithaca, NY, USA
4 Department of Medicine, Genetics and Molecular Pharmacology, Albert Einstein College of Medicine, Bronx, NY, USA
5 Cincinnati Children's Hospital Medical Center, Cincinnati, OH, USA
6 Department of Veterinary and Biomedical Sciences, Penn State College of Agricultural Sciences, University Park, PA, USA
7 Institute of Experimental Endocrinology, Biomedical Research Center, Slovak Academy of Sciences, Bratislava, Slovak Republic
8 The Department of Biology, Johns Hopkins University, Baltimore, MD, USA
9 Department of Chemistry, University of North Carolina, Chapel Hill, NC, USA
10 The Department of Pathology, New York University School of Medicine, New York, NY, USA
11 Department of Pathology, Albert Einstein College of Medicine, Bronx, NY, USA
12 Department of Microbiology and Immunology, Drexel University College of Medicine, Philadelphia, PA, USA
13 Department of Chemistry, City University of New York-Hunter College, New York, NY, USA
14 Molecular Biophysics Unit, Indian Institute of Science, Bangalore, India
15 Department of Pharmacy—Pharmaceutical Sciences, University of Bari Aldo Moro, Bari, Italy
16 Department of Environmental and Occupational Health Sciences, University of Washington, Seattle, WA, USA
17 Center for Metabolic and Vascular Biology, College of Health Solutions, Arizona State University, Scottsdale, AZ, USA
18 City University of New York, City College and Graduate Center, New York, NY, USA
19 Department of Pediatrics and Immunology, University of Connecticut, Farmington, CT, USA
20 Department of Pharmaceutical Sciences, University of Connecticut, Storrs, CT, USA
21 Department of Physiology and Pharmacology, University of Calgary, Calgary, AB, Canada
22 Division of Gastroenterology and Hepatology, Department of Medicine, Center for Gastrointestinal Biology and Disease, University of North Carolina at Chapel Hill, Chapel Hill, NC, USA
*Corresponding author. Tel: +420 585634903; E-mail: moulin@email.cz
**Corresponding author. Tel: +1 215 991 8135; Fax: +1 215 848 2271; E-mail: sandhya.kortagere@drexelmed.edu
***Corresponding author. Tel: +1 718 430 2871; Fax: +1 718 430 8550; E-mail: sridhar.mani@einstein.yu.edu
†These authors contributed equally to this work
‡Present address: St. Edmund's College, Shillong, Meghalaya, India
§Correction added on 17 March 2020, after first online publication: the author name has been corrected.

**Subject Categories** Pharmacology & Drug Discovery; Digestive System;
Immunology
February 2020 | Accepted 7 February 2020 | Published online 10 March 2020

## Introduction

Microbial metabolite mimicry serves as a means to probe areas of chemical space that have been previously underexploited for drug or probe discovery (Saha *et al*, 2016). However, to date, there has been no critical proof of this concept with regard to new drug discovery for the treatment of human disease.

Our laboratory had previously demonstrated that bacterial metabolism of L-tryptophan results in the formation of indoles and indole-derived metabolites, such as IPA (indole-3-propionic acid), which are active ligands for both mouse and human (h) pregnane X receptors (PXRs). Indoles regulate host small intestinal immunity *via* the TLR4 (Toll-Like receptor 4) pathway (Venkatesh *et al*, 2014). The use of PXR as a therapeutic target for inflammatory bowel disease (IBD) is a compelling concept in both rodents and humans (Cheng *et al*, 2012b). While a variety of PXR-activating xenobiotics exist, they all have significant off-target effects *in vivo* (e.g., drug interactions), which limits their use as clinical drugs for IBD (Cheng *et al*, 2012a).

Using a model developed in our laboratory (Venkatesh *et al*, 2014), we have shown that the co-occupancy of indole and IPA at the human PXR ligand-binding pocket or domain (LBD) is feasible. There are several opportunities to improve ligand binding *via* alteration of the pharmacophore H-bonding and pi-pi interactions (Venkatesh *et al*, 2014). The observation of PXR residue interactions with indole and IPA led us to propose that endogenous metabolite mimicry could yield more potent PXR ligands (agonists). Compounds derived from metabolite mimicry would have limited toxicity since they would derive from non-toxic metabolite-like subunits (Brave *et al*, 2015; Mani, 2017). Further support for this concept comes from the fact that basal indole levels in feces are in the low millimolar concentrations, and these levels are relatively non-toxic to mammalian cells ($LD_{50}$ oral in rats ~ 1 g/kg, Thermo Fisher Scientific Safety Sheet data) (Chappell *et al*, 2016). Similarly, indole metabolites like IPA, in feces, are likely to be present in low-to-high micromolar concentrations. However, the clinical utility of these concepts in drug or probe discovery remains elusive (Saha *et al*, 2016).

The goal of the present work was to establish the proof of concept that by developing mimics of microbial indoles using limited chemical analog development, one can significantly improve binding of indole derivatives to the receptors involved in abrogating intestinal inflammation.

## Results

### Characterization of FKK compounds as hPXR, AhR (aryl hydrocarbon receptor), and dual hPXR/AhR agonists

We have previously shown that indole and IPA can synergistically activate hPXR and produce functional effects *in vivo* (Venkatesh

*et al*, 2014; Ranhotra *et al*, 2016). Here, we hypothesized that the development of a small molecule, representing the interactions of both an indole and IPA, would be an innovative and potentially promising strategy for generating therapeutics targeting PXR in diseases such as IBD. To design such molecules, we used our platform technology called the hybrid structure-based (HSB) method. The HSB method utilized the interactions of both IPA and indole in the ligand-binding domain (LBD) of PXR. We screened the resulting pharmacophore, and the molecules ranked by their docking score. Two commercial molecules FKK999 and BAS00641451(BAS451) (Fig EV1A), which had docking scores of 65.89 and 52.66, respectively, were chosen as the starting point for further optimization. Docking studies demonstrated that although both FKK999 and BAS451 oriented in the binding pocket of PXR to maximize their interactions with the residues from LBD, FKK999 had a better interaction profile, resembling the interactions of indole and IPA in the same site (Fig EV1B). Docking of BAS451 shows several shared interactions with those of FKK999, but does not include the key ring stacking interaction with Trp299 and electrostatic interactions that contribute to the binding efficacy, since BAS451 has only two indole rings (and the additional phenyl ring does not compensate for the lost interactions). In PXR cell-based transactivation screens, FKK999 activated PXR. We used FKK999 for further design of improved bis-indole analogs (FKK 1-10), and we tested all analogs as PXR ligands using *in silico* approaches. As demonstrated by the interactions of FKK5 in the LBD (Fig EV1C), the ligand has arene-H interactions with Ser247 and Met250, electrostatic interactions with Met250, and Cys301 and other favorable interactions with Gln285, His407, Cys284, Met246, and Leu411.

We have reported the complete syntheses of indole metabolite mimic FKK1–FKK10 in Appendix Supplementary Methods and Results section. We also obtained crystal data and structures for lead compounds FKK5 (Table EV1; Fig EV1D) and FKK6 (Table EV2; Fig EV1E). The ligand efficiency metric (LEM) (Cavalluzzi *et al*, 2017) analysis (Abad-Zapatero & Blasi, 2011) for the FKK compounds predicted FKK5 and FKK6 as the best candidates for further studies. The same LEM analysis also indicated FKK4 as an efficient agonist; however, this compound could show metabolic instability with its *N*-protecting group possibly generating toxic aldehydes by oxidative cleavage *in vivo*.

All compounds synthesized were tested for their potential to activate hPXR, AhR, or both hPXR and AhR via luciferase assays as previously described (Goodwin *et al*, 1999; Huang *et al*, 2007; Novotna *et al*, 2011). All FKK compounds demonstrated a concentration-dependent effect on PXR activation (Fig 1A). By contrast, only FKK2 and FKK9 at 10 μM were biologically significant (> 100-fold) for AhR activation compared with dioxin (TCDD) control ligand. To a much lesser extent, we observed variable degrees of dose-dependent AhR activation profiles for FKK3, FKK4, FKK7, FKK8, FKK10, and FKK999 (Fig 1B). FKK compounds did not activate GR (glucocorticoid receptor) (Fig EV2A), VDR (vitamin D receptor) (Fig EV2B), TR (thyroid receptor) (Fig EV2C), and AR (androgen receptor) (Fig EV2D) using cell-based luciferase assays as previously described (Novotna *et al*, 2012; Bartonkova *et al*, 2015, 2016; Illes *et al*, 2015). Similarly, FKK5 or FKK6 did not induce constitutive androstane receptor (CAR) activity (Fig EV2E). Since RXR is an obligatory protein partner in the active PXR transcription complex, we also assessed the potential for FKK compounds to serve as ligands for RXR (Xie *et al*, 2000). Employing a

recently developed radio-ligand-binding assay for RXR (Toporova et al, 2016) on nuclear extracts from rat liver, we observed significant displacement of [³H]-9-cis-retinoic acid by FKK1 only, and to a lesser extent by FKK8 (Fig EV2F). Since FXR and PPARγ are nuclear receptors known to abrogate colitis (Ning et al, 2019), we employed a TR-FRET-based ligand displacement assay (FXR) (Fig EV2G) and a cell line reporter assay (PPARγ) (Fig EV2H). We did not observe any significant ligand agonist activity for both FKK5 and FKK6. While FKK6 activated human PXR, it did not significantly activate mouse PXR (Fig EV2I). Together, molecular docking, LEM analysis, PXR, and AhR transactivation assays showed that among the FKK compounds, FKK5 and FKK6 were the best chemical PXR-specific ligand leads.

## Characterization of FKK compound gene expression assay profile in cells

PXR agonists transcriptionally induce (> 2-fold) canonical target genes encoding drug metabolism enzymes/transporter, *CYP3A4* and *MDR1*, in both liver (hepatocytes) (Kandel et al, 2016) and intestinal cells (LS180) (Gupta et al, 2008). HepaRG cells simulate hepatocytes in that PXR ligands can also induce target genes in a similar but not identical manner (Aninat et al, 2006; Andersson, 2010; Antherieu et al, 2012). AhR agonists transcriptionally induce target genes, *CYP1A1* and *CYP1A2*, in both hepatocytes (Pastorkova et al, 2017) and intestinal cells (LS180) (Kubesova et al, 2016). As shown in Fig 1C (top panel) (see Materials and Methods, Table 1), in LS180 cells, FKK5 and FKK6 did not induce CYP1A1 mRNA. In PXR-transfected LS180 cells, FKK6 (and to a lesser extent FKK5) induced CYP3A4 (Fig 1C middle panel) and MDR1 mRNA (Fig 1C, bottom panel). In HepaRG cells harboring loss of PXR or AhR, there is diminished target gene induction when compared to the wild-type control cell line (Williamson et al, 2016; Brauze et al, 2017). TCDD (2,3,7,8—tetrachlorodibenzo-p-dioxin), a known AhR ligand, induces CYP1A1 mRNA in HepaRG™ control 5F clone and PXR-KO (PXR-knockout) cells. FKK5 and FKK6 did not induce CYP1A1 mRNA (Fig 1C, top panel). Rifampicin, a canonical PXR ligand, did not induce CYP3A4 mRNA in HepaRG™ PXR-KO cells compared to HepaRG™ control 5F clone or HepaRG™ AhR-KO cells. FKK6 and FKK5 induced CYP3A4 mRNA only in HepaRG™ control 5F clone and HepaRG™ AhR-KO cells, respectively (Fig 1C, middle panel). By contrast, and unlike in LS180 cells, none of the compounds, including rifampicin, induced MDR1 mRNA (Fig 1C, bottom panel). In primary human hepatocytes, TCDD induced CYP1A1 mRNA in all cells, whereas FKK5 and FKK6 did not (Fig 1C, top panel). Rifampicin and FKK6, unlike FKK5, showed more than twofold induction in one of four hepatocyte samples (Fig 1C, middle panel). Similarly, FKK6, but not FKK5, showed MDR1 mRNA induction in one of four hepatocyte samples (Fig 1C, bottom panel).

PXR target gene expression profile in the different cell-based models (Fig 1C) revealed a higher induction of CYP3A4 and MDR1 mRNA in LS180 cells when compared to HepaRG cells or primary human hepatocytes.

## Characterization of FKK compound cell and tissue cytotoxicity potential

The LS180 cell line was used to assess cell cytotoxicity using the MTT (3-(4,5-dimethylthiazolyl-2)-2,5-diphenyltetrazolium bromide)

assay as previously described (Bartonkova et al, 2015, 2016). The results show that the entire series of FKK compounds (up to 10 μM) did not significantly alter LS180 cell survival (Fig EV3A). Since *in vitro* cytotoxicity studies do not reflect *in vivo* effects, we conducted an acute toxicity study in C57BL/6 mice. FKK6 was selected as a lead compound for assessment, given that it had the more favorable PXR-selective ligand activity in cells. A dose of 500 μM in 10% DMSO (dimethyl sulfoxide) (Caujolle et al, 1967a; Castro et al, 1995) per day was chosen since as it represented the maximal solubility of FKK6 in an aqueous solution that could be safely administered to mice. The mice were gavaged for ten consecutive days and necropsied on day 12 with the collection of blood (serum). FKK6 did not alter the serum chemistry profiles assessed (Fig EV3B). FKK6 did not significantly impact tissue pathology when compared to control (vehicle)-exposed mice (Table EV3). Similarly, a 30-day toxicity test was conducted in mice using FKK6 at an oral dose of 200 μM in 0.8% DMSO per day. We chose this dose to stay within the safe DMSO dose administered to mice over 30 days. On day 30, we necropsied the mice and collected blood (serum). FKK6 did not significantly alter serum chemistry profiles (Fig EV3C and D) or impact tissue pathology when compared to control (vehicle)-exposed mice (Table EV4).

## Characterization of FKK-induced PXR–DNA interactions via chromatin immunoprecipitation (ChIP)

To determine whether FKK5 and FKK6 could enhance occupancy of specific PXR DNA-binding elements (PXRE) in cells, we performed ChIP assays using LS174T cells transiently transfected with a PXR-expressing plasmid. Extensively passaged LS174T cells endogenously express PXR but with insufficient levels for PXR protein pull-down and reporter experiments (Delfosse et al, 2015). PXR transfection allowed us to isolate PXR occupancy effects in response to ligands confidently. We verified a semi-quantitative ChIP using PXR-transfected LS174T cells exposed to FKK6 or its vehicle (DMSO). FKK6 augmented PXR occupancy of the proximal CYP3A4 and MDR1 promoter (at their respective, PXR-binding elements) (Fig 1D) (see Materials and Methods, Table 3).

## Characterization of FKK5 and FKK6 as direct ligands of PXR in solution

Kinase-dependent phosphorylation (e.g., cdk2) represses PXR activation (Smutny et al, 2013). We tested FKK5 and FKK6, and neither of these compounds inhibited the kinases tested *in vitro* (DiscoverX scanMAX^SM assay panel of 468 kinases) (Fig EV4A and Table EV5). Together, these results support a direct interaction of FKK compounds with PXR. To demonstrate this interaction, we performed a cell-free competitive hPXR TR-FRET-binding assay. In this assay, FKK5 demonstrated an IC₅₀ of 3.77 μM, similar to FKK6 ~ 1.56 μM (Fig 1E). To obtain direct proof of interaction with hPXR-LBD, we performed isothermal titration calorimetry (ITC) experiments using His-tagged hPXR-LBD protein (PXR.1) in solution. ITC measures the affinity, $K_a$, and Gibbs energy ($\Delta G = -RT\ln K_a$) and the changes in enthalpy, $\Delta H$, and entropy, $\Delta S$, associated with the binding of the FKK compounds ($\Delta G = -RT\ln K_a = \Delta H - T\Delta S$). Enthalpic and entropic contributions to binding affinity guide drug design and optimization (Garbett & Chaires, 2012). In previous

work, we demonstrated that IPA alone has weak hPXR-LBD interactions as determined using cell-based assays (Venkatesh *et al*, 2014). Consistent with this observation, IPA did not induce a thermal signature concerning its interaction with the hPXR-LBD in solution in our ITC assay, unlike the canonical hPXR agonist rifampicin (Fig EV4B). FKK5 bound to a single site in LBD, albeit with better affinity than IPA ($K_d = 0.30$ μM) due to a more favorable enthalpy of binding ($\Delta H = -2.5$ kcal/mol, $-T\Delta S = -6.4$ kcal/mol) (Fig 1F). Compared to FKK5, FKK6 bound with improved enthalpy ($\Delta H = -3.5$ kcal/mol) but with an even larger loss in favorable entropy ($-T\Delta S = -4.1$ kcal/mol), which translated to a loss in binding affinity for FKK6 ($K_d = 2.7$ μM) (Fig 1F).

We have previously shown that unlike rifampicin, indole in combination with IPA is unable to activate the PXR mutant (C285I/C301A) (Venkatesh *et al*, 2014). Similarly, to test the effect FKK6 (10 μM) on this mutant, 293T cells were transfected with wild-type PXR.1 or mutant PXR.1 (C285I/C301A) plasmids along with its cognate luciferase reporter. We exposed transfected and control cells to DMSO, rifampicin (10 μM), or FKK6 (10 μM) and assessed the reporter activity. While rifampicin significantly induced PXR reporter activity in both wild-type and mutant PXR-transfected cells, FKK6 was unable to activate mutant PXR (Fig 2A). Together, these data show that FKK5 and FKK6 are *bonafide* agonist ligands of PXR.1 in solution and cells.

### Characterization of FKK5 and FKK6 on PXR-TLR4 and NF-κB signaling in human colon cancer cells

We have previously shown that PXR activation down-regulates TLR4 mRNA expression, leading to the inhibition of NF-κB signaling (Venkatesh *et al*, 2014; Mani, 2016a; Ranhotra *et al*, 2016). We have also shown that FKK5 and FKK6 directly interact with and activate PXR-LBD. Thus, to determine the effect of FKK5 and FKK6 on the PXR-TLR4-NF-κB signaling pathway, we used PXR-transfected Caco-2 cells. Undifferentiated Caco-2 cells inherently have low PXR expression (not detected) but relatively higher TLR4 expression and are excellent *in vitro* models for the study of enterocyte function (Delie & Rubas, 1997; Mani, 2016a; Ranhotra *et al*, 2016; Hung & Suzuki, 2018). FKK5 and FKK6 down-regulated TLR4 (0.39-fold and 0.49-fold, respectively) but up-regulated CYP3A4 (3.7-fold and 3.29-fold, respectively) and MDR1 (5.12-fold and 6.45-fold, respectively) mRNA expression (Fig 2b) (see Materials and Methods, Table 2). Together, these results provide a rationale for the study of FKK compounds on modulating the PXR-TLR4-NF-κB pathway.

To determine whether FKK5 and FKK6 also inhibit NF-κB signaling in intestinal cells, we used PXR and NF-κB reporter-co-transfected and NF-κB reporter-transfected LS180 intestinal cell line in early and late passages. In this assay, we verified the enhanced expression of PXR protein by immunoblotting (Fig EV4C). FKK5 and FKK6 significantly reduced in a dose-dependent manner TNF-α-induced NF-κB reporter activity in PXR-untransfected and PXR-transfected LS180 cells (Fig 2C). The effects of FKK compounds on inhibition of TNF-α-induced NF-κB reporter activity were significantly greater in PXR-transfected cells as compared to PXR-untransfected LS180 cells (Fig 2C). Notably, the trend for effects of FKK5 and FKK6 was similar throughout LS180 cell line passages (Fig 2C and D). Notably, PXR-transfected LS180 cells induced PXR target

genes (*CYP3A4*, *MDR1*) well above untransfected cells, and FKK5 and FKK6 induced PXR target gene expression in the transfected LS180 cells (Fig EV4D).

To reproduce these findings in a more robust model, we used LS174T cells, in which PXR protein was deleted via a CRISPR/Cas9 knockout of the human PXR locus. The LS174T PXR-knockout cells (*PXR*-KO or *NR1I2* KO) are pooled transfectants (editing efficiency ~ 83%). These cells gained functional PXR activity with multiple passages (Fig EV5A) even though the PXR protein expression remained very low to undetectable (Fig EV5B). FKK5 and FKK6 did not induce PXR target genes in *PXR*-KO cells (Fig EV5C). In early passage *PXR*-KO cells, we used PXR and NF-κB reporter-co-transfected and NF-κB reporter-transfected *PXR*-KO cell line. However, we were unable to see sustained PXR protein in the transiently transfected *PXR*-KO cells using a PXR plasmid. Thus, we proceeded with establishing the effect of rifampicin and FKK6 in the TNF-exposed *PXR*-KO cell line transfected with the NF-κB reporter. Unlike in wild-type LS174T cells, neither rifampicin nor FKK6 repressed NF-κB reporter activity (Fig 2E). Together, these data confirm that FKK5 and FKK6 are agonist ligands of PXR and that they directly repress TLR4-TNF-α-NF-κB in varied intestinal cells via PXR.

### Characterization of FKK5 and FKK6 in human colon cancer cells and intestinal organoids exposed to inflammatory and infectious stimuli

To further characterize the anti-inflammatory effects of FKK5 and FKK6 on human intestinal cells and tissue, we first determined their effects on Caco-2 cell monolayers. These cells are well-established models to study intestinal cell differentiation and physiology, specifically under conditions of inflammatory stimuli (Hung & Suzuki, 2018). Differentiated Caco-2 cells following long-term culture have increased PXR expression (Mani, 2016a). In these cells, TNF-α significantly induced IL-8 mRNA; however, only FKK6 (25 μM) significantly reduced cytokine-induced IL-8 (Fig 3A). Interestingly, cytokines also considerably increased salmonella invasion in Caco-2 cells when compared to untreated cells (Fig 3B). FKK5 and FKK6, at all concentrations tested (10 and 25 μM), and compared to cytokines only, significantly decreased cytokine-induced salmonella invasion (Fig 3B). No significant effect of FKK5 or FKK6 alone was observed on salmonella invasion (Fig 3B). Similarly, in Caco-2 cells, NF-κB is a seminal regulator of both IL-8 and other pro-inflammatory cytokines. In Caco-2 cells, at 2 or 12 h, cytokines (CK only) translocated most of the NF-κB signal by immunofluorescence from the cytoplasm to the nucleus (when compared to No CK) (Fig 3C). In the presence of FKK5 or FKK6 (25 μM), an increased cytoplasmic NF-κB signal was observed, suggesting more retention of NF-κB outside the nucleus (Fig 3C).

While Caco-2 cultures are relevant models to study intestinal barrier function, culture conditions can greatly impact the experimental results (Delie & Rubas, 1997). Thus, we investigated the impact of our FKK analogs on primary human intestinal organoids (HIOs). In iPSC-derived HIOs, cytokines induced a significant increase in IL-8 mRNA levels at 12 h (Fig 3D; $P < 0.05$, two-way ANOVA) but not at 2 h after exposure. FKK5 and FKK6 (10 and 25 μM, respectively) had no significant effect on basal IL-8 expression (Fig 3D); however, in the presence of cytokines, both FKK5

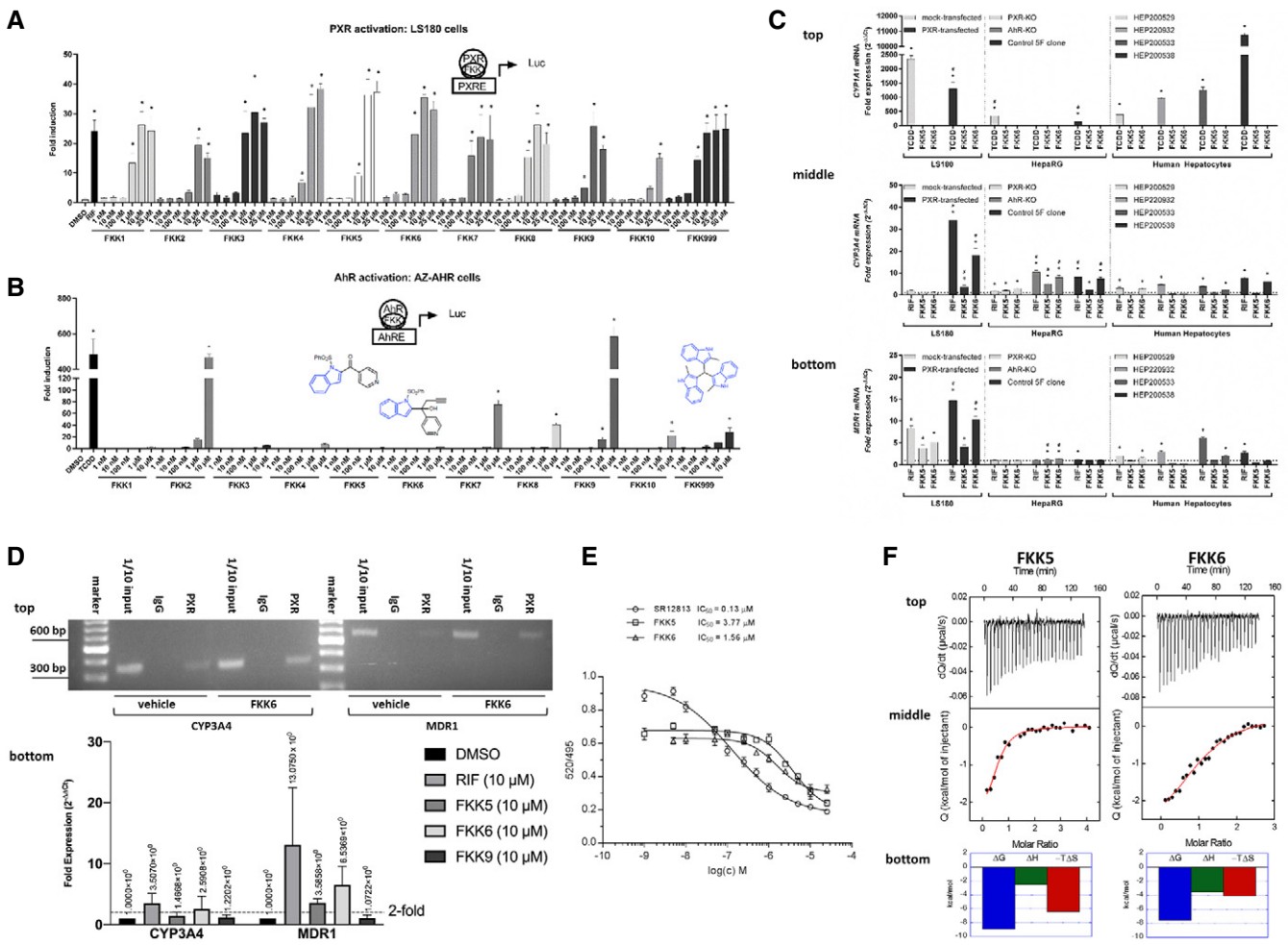

**Figure 1. FKK5 and FKK6 activate PXR.**

A, B  (A) Histogram (mean, 95% CI) of fold induction of PXR activity reporter assay (luc, luciferase) in LS180 cells transiently transfected with wild-type PXR and p3A4-luc reporter plasmids. (B) Same as (A) in HepG2 cells stably expressing pCYP1A1 luciferase plasmid (AhR reporter). Chemical structures of FKK5, FKK6, and FKK999 compounds are overlaid. Indole is colored blue. (A, B) The bar graph(s) depicts one representative experiment of a series of experiments ($n > 3$) performed in four consecutive passages of cells. *$P < 0.05$, one-way ANOVA with Dunnett's *post hoc* test. *Significant over vehicle (DMSO) control.

C  Histogram (mean, 95% CI) of fold mRNA expression, CYP1A1 (top panel), CYP3A4 (middle panel), and MDR1 (bottom panel) in LS180 cells with or without (mock) transfected PXR plasmid, HepaRG hepatic progenitor cells (PXR-knockout, PXR-KO; AhR-knockout, AhR-KO; parental control 5F clone), and primary human hepatocytes (HEP) from four donors is shown. The bar graph represents one experiment of a series of experiments ($n > 3$) performed in four consecutive passages of LS180 cells; $n = 2$ independent experiments with one well/compound and RT–PCR performed in triplicate for each HepaRG genotype; for each donor hepatocyte, $n = 1$ well/compound and RT–PCR performed in triplicate. *,#$P < 0.05$, two-way ANOVA with Tukey's *post hoc* test. *Significant over vehicle control. #Significant over the same treatment in corresponding mock-transfected or knockout cells.

D  Chromatin immunoprecipitation (ChIP) assay in LS174T cells. Top panel, PCR products cells exposed to vehicle or FKK6 and run on a 2% agarose gel. DNA 100 base pair marker; vehicle, 10% DMSO; FKK6 (10 μM); 1/10 input—0.2 million cells before IP; IgG—IP with polyclonal rabbit IgG; PXR—IP with PXR antibody. Bottom panel, quantitative PCR from the ChIP assay for compounds tested with the gene-specific PCR amplicon normalized to GAPDH (fold expression). Dash line, twofold expression. The data are one representative experiment of two independent experiments (each $n = 3$ biological replicates, $n = 4$ technical replicates). Mean fold expression is expressed above each histogram (mean ± SD).

E  PXR TR-FRET assay. TR/FRET ratio (520/495 nm) is plotted against concentration of compound(s). Half-maximal inhibitory concentrations IC50 were obtained from interpolated standard curves (sigmoidal, 4PL, variable slope); error bars show standard deviation of n = 2 independent experiments each with four technical replicates.

F  Microcalorimetric titrations of PXR ligand-binding domain (LBD) with FKK5 and FKK6. A single-site binding model was used for fitting the data. The upper panel shows the output signal, $dQ/dt$, as a function of time. The middle panel shows the integrated heats as a function of the ligand/PXR molar ratio in the cell. The solid line represents the best non-linear least-squares fit of the data. The lower panel shows respective thermodynamic signatures of binding to the PXR ligand-binding domain.

Source data are available online for this figure.

and FKK6 significantly reduced IL-8 mRNA expression after 12 h of exposure (Fig 3D). Similarly, cytokines significantly increased salmonella invasion in HIOs (Fig 3E). FKK5 and FKK6 (10 and 25 μM, respectively) had no significant effect on basal levels of salmonella invasion (Fig 3E); however, when compared to cytokines alone, both FKK5 and FKK6 significantly reduced salmonella

**Table 1. Primer sets.**

| Gene symbol | Forward primer sequence | Reverse primer sequence | UPL probe No. |
| --- | --- | --- | --- |
| CYP1A1 | CCAGGCTCCAAGAGTCCA | GATCTTGGAGGTGGCTGCT | 33 |
| CYP1A2 | ACAACCCTGCCAATCTCAAG | GGGAACAGACTGGGACAATG | 34 |
| CYP3A4 | TGTGTTGGTGAGAAATCTGAGG | CTGTAGGCCCCAAAGACG | 38 |
| MDR1 | CCTGGAGCGGTTCTACGA | TGAACATTCAGTCGCTTTATTTCT | 147 |
| GAPDH | CTCTGCTCCTCCTGTTCGAC | ACGACCAAATCCGTTGACTC | 60 |

invasion (Fig 3E). Interestingly, salmonella invasion was significantly reduced at all time points studied (2, 4, 6, and 12 h post-exposure to cytokines). In HIOs, at 2 or 12 h, cytokines (CK only)

translocated most of the NF-κB signal from the cytoplasm to the nucleus (CK+FKK compared to no CK) (Fig 3F). In the presence of FKK5 or FKK6 (25 μM), however, there was an increased

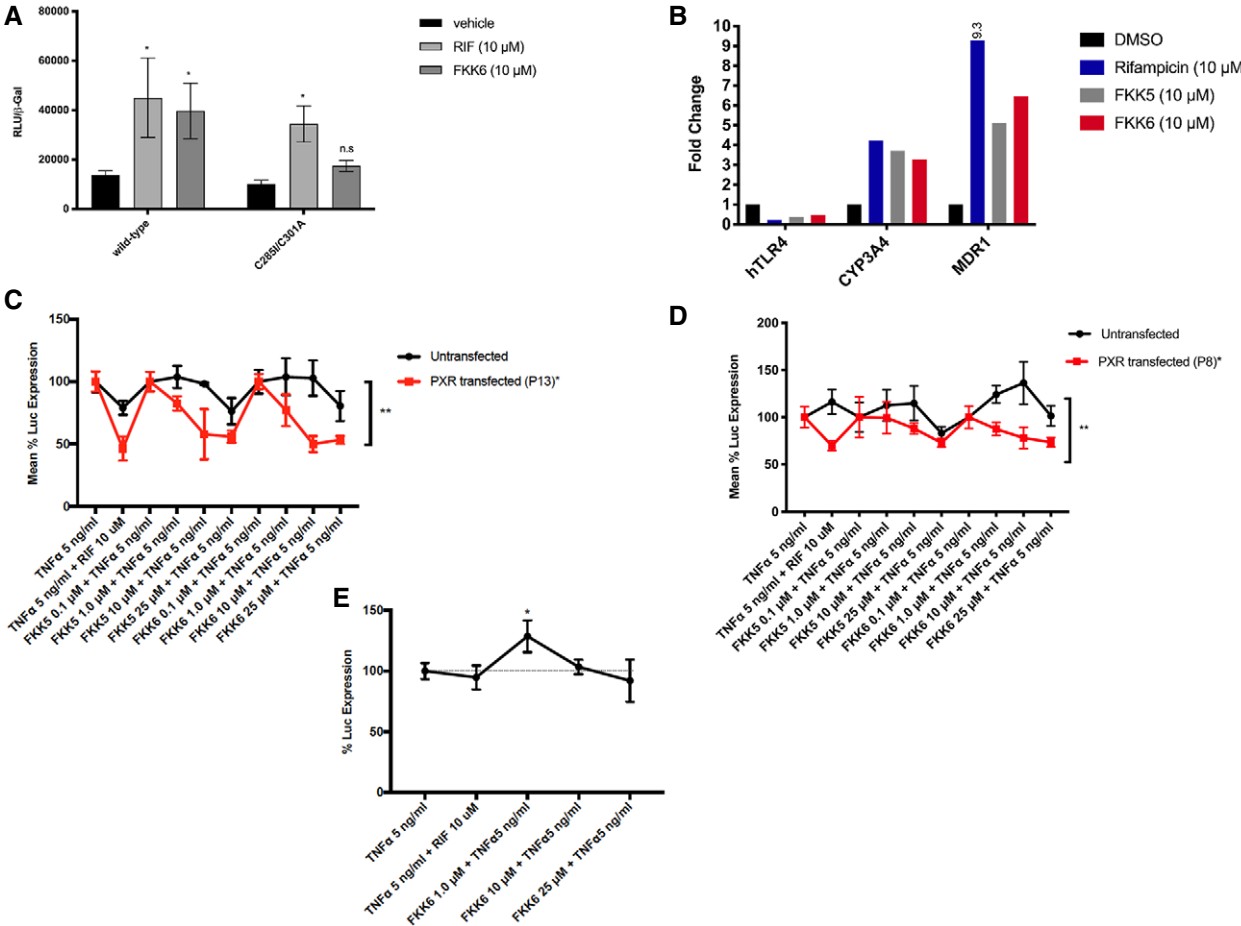

**Figure 2. FKK5 and FKK6 inhibit NF-κB activation in a PXR-dependent manner.**

A  PXR (luciferase) reporter assay in HEK293T cells transiently transfected with PXR plasmids (wild-type and ligand-binding domain mutant C285I/C301A). RLU, relative light units, are shown normalized to beta-galactosidase (β-Gal) expression. The histogram represents one experimental data (of $n > 2$ independent experiments each performed in quadruplicate): mean (95% CI); n.s. not significant; *$P < 0.05$, two-way ANOVA.

B  Histogram represents fold change in mRNA expression, normalized to GAPDH, by RT–qPCR from Caco-2 cells exposed to compounds. DMSO, 0.1% DMSO vehicle; 9.3 is 9.3-fold expression. The data are one representative experiment of two independent experiments (each $n = 3$ biological replicates, $n = 4$ technical replicates).

C–E  Points and error represent mean (95% CI) values for control (TNF-α, 5 ng/ml) normalized NF-κB (luciferase) reporter activity in LS180 cells after (C) Passage 13 (P13) and (D) Passage 8 (P8), and in (E) LS174T PXR-KO (knockout) cells transiently transfected with pNL3.2.NF-κB-RE vector. (C, D) Line graph depicts one representative experiment of a series of experiments ($n > 4$) performed in four consecutive passages of cells. Data were expressed as mean with 95% CI of four technical replicates. (E) Mean (± SD). *,**$P < 0.05$, two-way ANOVA with Tukey's multiple comparison test.

Source data are available online for this figure.

cytoplasmic NF-κB signal, suggesting more retention of NF-κB outside the nucleus (Fig 3F).

In another HIO system, we confirmed that the analogs FKK5 and FKK6 significantly induced PXR target genes (Appendix Fig S1A; see Materials and Methods, Table 4). Interestingly, CYP3A4, a target gene that is only modestly (as opposed to MDR1) elevated in intestinal cell lines exposed to FKK5 (Fig 1C), was also only slightly increased in the HIO experiments (Appendix Fig S1A: borderline significance for CYP3A4, $P = 0.056$). To determine whether FKK compounds inhibit TNF-α-mediated induction of pro-inflammatory cytokine (IL-8), HIOs from different individuals were exposed to TNF-α in the presence or absence of FKK5 ($n = 9$) or FKK6 ($n = 3$) or FKK9 ($n = 5$). In a pair-wise comparison of HIOs from the same individual exposed to DMSO or FKK compounds, FKK5 (10 μM) significantly lowered IL-8 mRNA levels (Fig 4A). Similarly, FKK6 (10 μM) reduced IL-8 mRNA levels in organoids; however, this decrease did not reach statistical significance ($P = 0.07$; Fig 4B). FKK9, which is not only a potent AhR agonist, but also has PXR agonist activity, did not show significant effects on IL-8 mRNA levels in this assay (Fig 4C).

Together, these data from Caco-2 cells and HIOs derived from multiple sources and centers suggest that the lead FKK compounds, FKK5 and FKK6, are capable of inducing PXR target gene expression in all intestinal cell types. The capacity to induce PXR target gene

expression also correlates with the reduction of TNF-α-induced pro-inflammatory cytokine (IL-8) expression, reduced invasion from intestinal pathobiont, *Salmonella typhimurium,* and reduced accumulation of nuclear NF-κB.

**PXR target gene expression in mice**

The FKK compounds presented here are first-generation indole/IPA analogs, not optimized for pharmaceutical discovery. However, to determine the effects *in vivo*, we performed studies with our representative best lead, FKK6, in C57BL/6 mice. FKK6 was gavaged at concentration ~ 500 μM in 10% DMSO and administered to C57BL/6 mice over 36 or 60 h (3 or 5 doses, respectively). No significant effect was observed on PXR target gene (cyp3a11, mdr1a, mdr1b, or mdr1) expression across organs (liver, and small and large intestines) in mice (Fig 4D: 36 h; Appendix Fig S1B: 60 h). Induction of cyp3a11 (2.07-fold) was observed in the large intestine of wild-type mice (Appendix Fig S1B). In $pxr^{-/-}$ mice exposed to FKK6, the target gene expression levels (cyp3a11, mdr1) were decreased ($\leq 1.5$-fold) compared to those of DMSO-exposed mice. There was no effect on PXR target gene (mdr1) expression across organs (liver, and small and large intestines) in $pxr^{-/-}$ mice (Fig 4E: 36 h; Appendix Fig S1C: 60 h). However, we show that cyp3a11 was significantly suppressed in the small (0.25× DMSO) and large

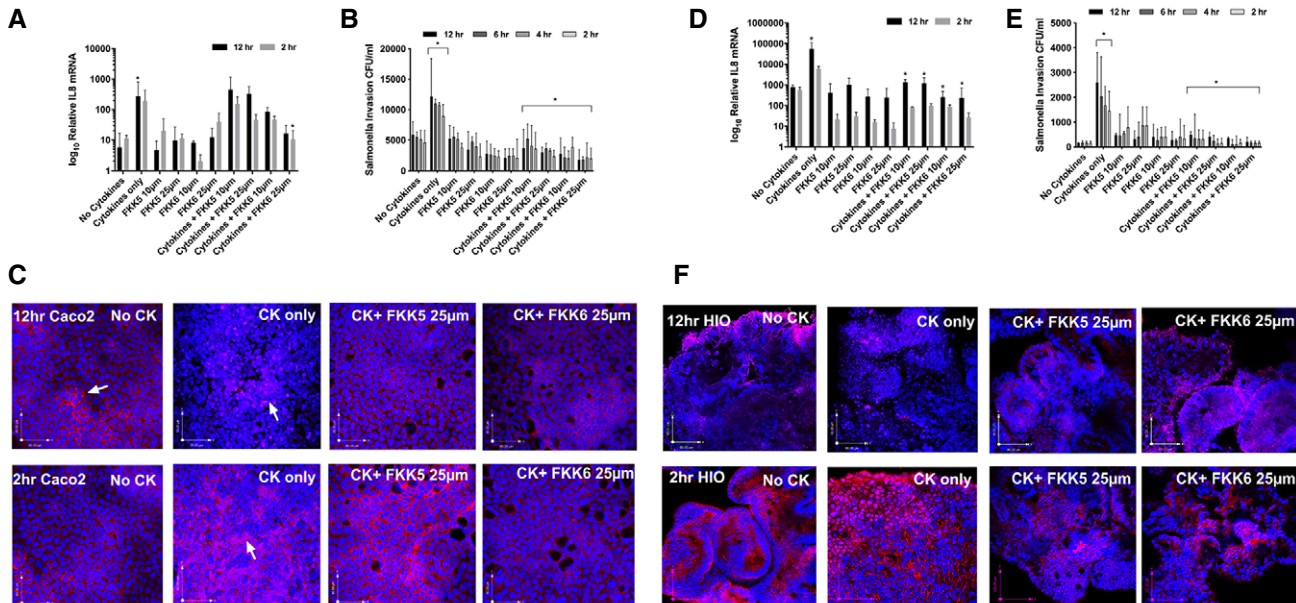

**Figure 3. FKK5 and FKK6 inhibit cytokine and salmonella infection-induced NF-κB nuclear translocation.**

A   In Caco-2 monolayer cells, $\log_{10}$ relative expression of IL-8 mRNA with or without cytokines (50 ng/ml IL-1, 10 ng/ml IFN-γ, 10 ng/ml TNF-α), and cytokine cocktail plus FKK drugs (10 and 25 μM) with 2-h or 12-h incubations.

B   Salmonella invasion in colony-forming units per ml (CFU/ml) at the different time periods of incubation.

C   Confocal images of NF-κB (red) and the nucleus (blue). Scale bars = 80 μm.

D–F   Same as (A–C) in human intestinal organoids (HIO). Scale bars ($x$–$y$, bottom left) = 70–100 μm.

Data information: The histogram, mean with 95% CI, depicts combined data from biological replicates ($n = 3$) with three consecutive passages of cells. Each biological replicate has three technical replicates. *$P < 0.05$, two-way ANOVA with Tukey's *post hoc* test. * for cytokines vs. no cytokines, compares either the 12, 6, 4 h, or 2 h equivalent time point(s) as indicated; * for cytokines vs. cytokines + FKK compound, compares either the 12, 6, 4 h, or 2 h equivalent time point(s) as indicated. (C) Arrows represent localization of NF-κB (red, cytoplasmic; purple, nuclear). Confocal images shown are representative images from $n = 3$ biological replicates.

Source data are available online for this figure.

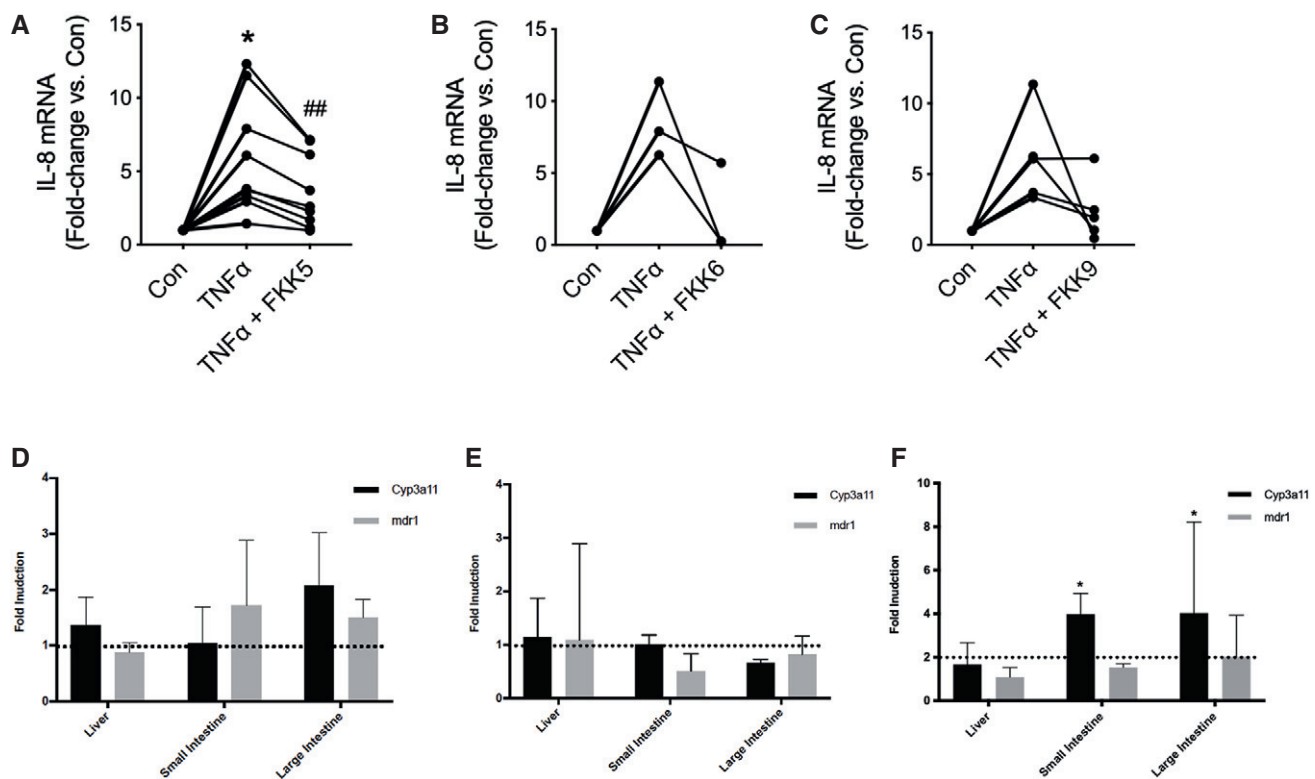

**Figure 4. FKK6 inhibits cytokine-mediated IL-8 induction in human colonic organoids and induces PXR target genes in mice.**

A–C Quantitative gene expression by RT–qPCR of IL-8 mRNA from human colonic enteroids (over 40 enteroids per individual sample per well) exposed to control (vehicle) or TNF-α with or without (A) FKK5 ($n = 9$ paired samples), (B) FKK6 ($n = 3$ paired samples; one removed due to poor inflammatory response), and (C) FKK9 ($n = 5$; 3 removed due to poor response and tissue integrity). Data shown as mean fold change relative to vehicle control (con) for each paired sample ($n = 3$ replicates). Each paired sample data set represents biopsies from an individual patient. *,##$P < 0.05$, pair-wise one-way ANOVA with Tukey's *post hoc* test.

D–F Fold induction (mRNA) of genes in (D) C57BL/6 mice, (E) *pxr*$^{-/-}$ mice, and (F) *hPXR* mice (mice expressing the human PXR gene) gavaged with vehicle (10% DMSO; $n = 3$) or FKK6 (500 μM in 10% DMSO; $n = 3$) every 12 h for 3 total doses. The entire experiment was repeated two independent times, and one representative experiment is shown. Each mouse (each organ) was studied in quadruplicate assays and normalized to internal control, GAPDH. The histograms show mean (95% CI) values for gene expression. The dotted line marks (D, E) basal fold expression and (F) twofold gene induction. *$P < 0.05$, two-way ANOVA with Tukey's *post hoc* test.

Source data are available online for this figure.

intestines (0.36× DMSO) at 60-h exposure (Appendix Fig S1C). These data suggest that PXR might be necessary in these organs as basal regulators of these genes. By contrast, there is a biologically significant induction of cyp3a11 (but not mdr1) expression in small and large intestines as compared to DMSO or liver cyp3a11 expression in *hPXR* mice (mice expressing the human PXR gene) when exposed to 36 h of FKK6 (three doses) (Fig 4F). Interestingly, when FKK6 administration was prolonged (FIVE treatments or 60-h exposure), a distinct increase in mdr1 expression was observed in the liver (as compared to the intestines), but no significant effects were measured on cyp3a11 expression in *hPXR* mice (mice expressing the human PXR gene) (Appendix Fig S1D). Together, these results demonstrate that the FKK compounds induce PXR target genes in a PXR-dependent manner in mice.

### Effect of FKK6 in mouse model of acute colitis *in vitro* and *in vivo*

To characterize the impact of a lead compound, FKK6, on murine colitis, adult C57BL/6 female *hPXR* mice (mice expressing the

human PXR gene) were acutely exposed to DSS. Mice were administered once daily vehicle/FKK6 (200 μM) by oral gavage plus intrarectal bolus for 10 days until necropsy. FKK6-treated mice showed significantly reduced weight loss and increased colon length on day 10 compared to vehicle-treated mice (Fig 5A and B). The inflammation score was also substantially decreased in FKK6-treated mice (Fig 5C); however, the fecal lipocalin 2 and FITC-dextran measurements revealed only a trend in the reduction (Fig 5D and E). We replicated these findings using the clustering method (weight loss, colon length), which correctly identified the two groups of mice. By including fecal lipocalin 2 as a cluster feature in the analysis, the clustering method identified the two groups of mice correctly except for one of the samples. The intestinal tissue cytokines, IL-10 and TNF-α (and TNFR2), showed a significant increase and decrease, respectively, in FKK6-treated mice compared with controls (Fig 5F; Materials and Methods, Table 5). IL-17f, an IL-17a heterodimer partner, is a prime mediator of colitis (Tang *et al*, 2018), and we showed a significant reduction in this cytokine expression levels in FKK6-treated mice

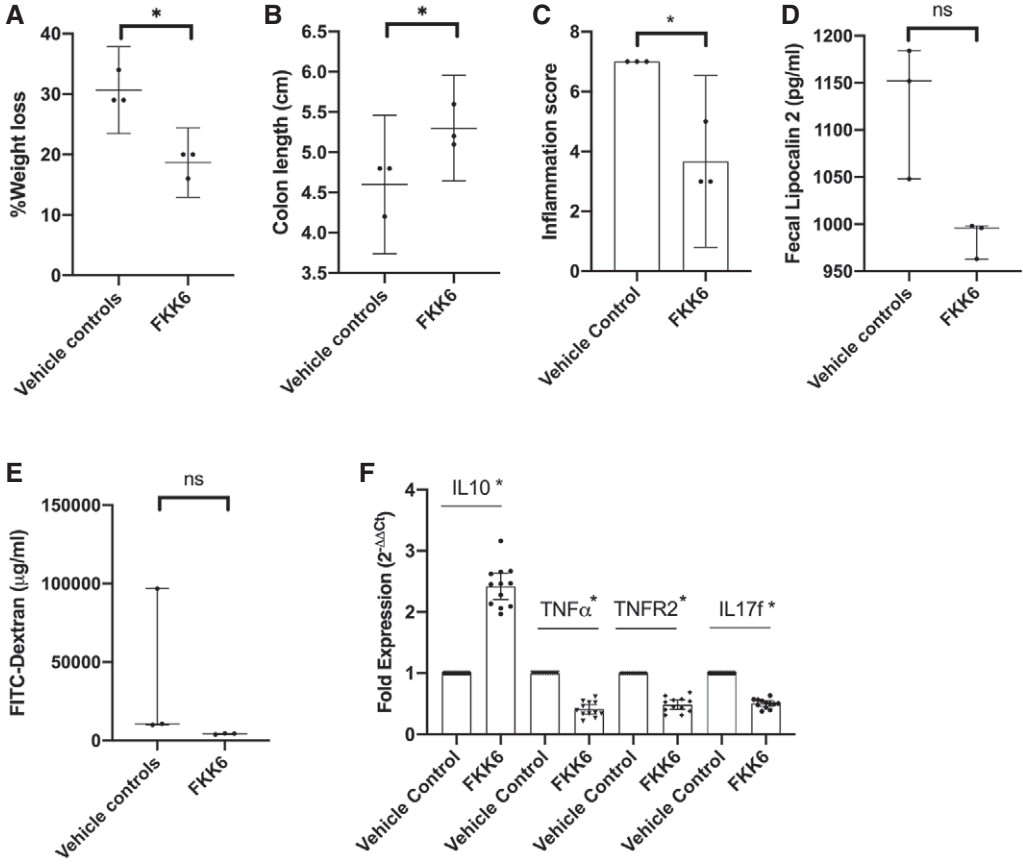

**Figure 5. FKK6 abrogates dextran sulfate sodium-induced colitis in mice expressing the human PXR gene.**

A–F  After coin-toss randomization, *hPXR* mice (mice expressing the human PXR gene) were allocated to treatment with vehicle (0.8% DMSO) (*n* = 3/genotype) or FKK6 (200 micromolar) (*n* = 3/genotype), by simultaneous oral gavage and intrarectal delivery starting day 1–10 of DSS administration. (A) % weight loss from baseline (day 1 vs. day 10), (B) colon length (cm), (C) inflammation score (see Materials and Methods), (D) fecal lipocalin 2 (pg/ml), (E) serum FITC-dextran (μg/ml), (F) fold expression of mRNA.

Data information: (A–C) Mean (95% CI). (D, E) Median (interquartile range). (F) Fold expression of mRNA as illustrated in *hPXR* mouse colon tissue exposed to DSS (*n* = 3/group; each PCR performed in quadruplicate). (A–E) The entire experiment was repeated three independent times, and one representative experiment is shown. *(A, B, C) *P* < 0.05, Welch's *t*-test; *(D, E) *P* < 0.05, Mann–Whitney test; ns, not significant; (F), * compares FKK6 vs. vehicle control for each gene, *P* < 0.05, two-way ANOVA. Source data are available online for this figure.

compared to vehicle-treated mice (Fig 5F). On the contrary, FKK6 did not alter weight loss, colon length, inflammation score, lipocalin 2, or FITC-dextran levels in *pxr⁻/⁻* mice (Fig 6A–E).

Together, these results show that a representative FKK lead compound, FKK6, significantly reduces dextran sodium sulfate (DSS)-induced colitis in mice in a PXR-dependent manner.

## Discussion

A longstanding problem in drug design and discovery is the limited region of chemical space exploited by the molecules conventionally available to small and large pharmaceuticals and academia. Others have proposed that natural metabolite derivatives are likely to serve as more potent and well-tolerated drugs; proof of this concept is lacking (Saha *et al*, 2016). Herein, we show that indole compounds of the FKK series mimic the docking of the natural indole and indole propionic acid derivatives for PXR, and have produced two potent lead compounds, FKK5 and

FKK6. While not optimized for therapeutic delivery, these leads showed significant activity in suppressing intestinal inflammation *in vitro* and, for FKK6, in mice and human tissues simulating colitis.

There are several caveats to bear in mind when evaluating the data presented. While we show the direct binding of FKK6 to PXR protein, our studies have not identified the direct binding residues that interact with FKK6. Research to obtain a crystal structure of co-crystallized PXR and FKK is currently underway. The *in vivo* disposition of FKK compounds and their metabolites is presently unknown. Pharmacokinetic profiling will help explain why FKK compounds do not activate PXR in liver cells (primary hepatocytes and HepaRG) compared to intestinal cells (LS180 and LS174T). The metabolites generated in mice may also inhibit parent compound interactions with PXR, which will be studied further. Recent research revealed that intestinal PXR activators can induce hypertriglyceridemia and metabolic perturbations (Meng *et al*, 2019). We did not observe any significant changes in lipid profiles in mice exposed to either a 10-day or 30-day FKK6 gavage. It is

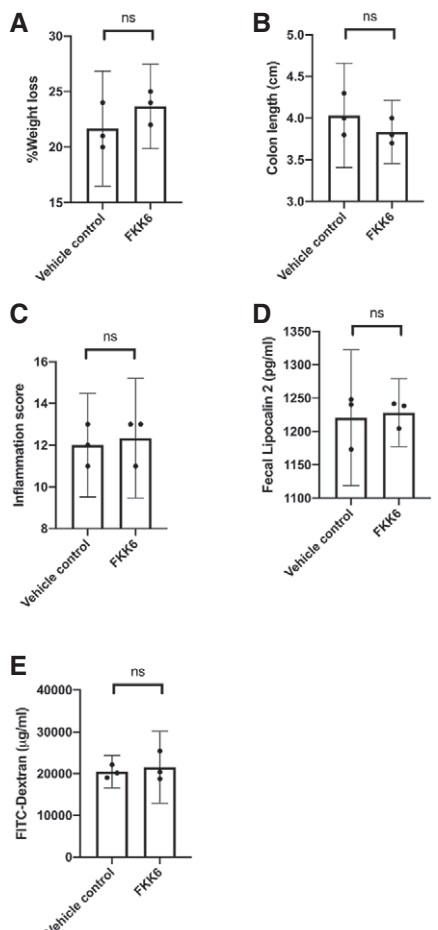

**Figure 6. FKK6 does not abrogate dextran sulfate sodium-induced colitis in *pxr*$^{-/-}$ mice.**

A–E As in Fig 5, *pxr*$^{-/-}$ mice underwent the identical experimental procedure and analysis. The entire experiment was repeated three independent times, and one representative experiment is shown.

Data information: (A–E) Mean (95% CI); Welch's *t*-test; ns, not significant. Source data are available online for this figure.

potential of these compounds. Future strategies to improve solubility and targeted delivery could include formulations used in existing compounds for colon-targeted delivery (Guo *et al*, 2018). Despite these caveats, our study provides and validates the initial proof of concept of microbial metabolite mimicry as a potential drug discovery modality.

The prototypical pharmacologic ligands of human PXR include rifaximin, rifampicin, hyperforin, and SR12813 (Orans *et al*, 2005). Rifaximin is a human PXR agonist ligand, but, as an antibiotic, changes intestinal microbiota, which results in the emergence of drug resistance upon chronic administration (Farrell, 2013). Furthermore, despite the claim that rifaximin is non-absorbable via the intestinal tract, chronic administration of rifaximin in mice resulted in hepatic steatosis (Cheng *et al*, 2012a). Indeed, while controversial (Abdel-Razik *et al*, 2018), rifaximin can potentially induce insulin resistance in humans and is not beneficial in the treatment of non-alcoholic steatohepatitis (Cobbold *et al*, 2018). Finally, for the rifamycin family (rifaximin and rifampicin), it is possible that targets other than nuclear receptors, such as p300 or other enhancer activation, may affect the drug's pleiotropic effects in tissues (Smith *et al*, 2014; Yan *et al*, 2017). Upon chronic dosing, rifampicin accumulation is not targeted toward the intestines but the brain and lungs (Shan, 2004). Similarly, hyperforin has multiple cellular targets (e.g., reactive oxygen species, elastases, Protein kinase C) (Feisst & Werz, 2004; McCue & Phang, 2008; Galeotti *et al*, 2010). Hyperforin pharmacokinetics suggest rapid absorption without accumulation in tissues and, thus, with each dose causing clinically relevant adverse drug–drug interactions (Biber *et al*, 1998; Soleymani *et al*, 2017). Hyperforin can induce reversible liver injury in mice (Negres *et al*, 2016). SR12813 induces degradation of 3-hydroxy-3-methylglutaryl-coenzyme A (HMG-CoA) reductase independent of effects on PXR (Berkhout *et al*, 1996). SR12813 was never clinically developed as an anti-cholesterol drug. Our PXR ligand, in particular, FKK6, tends to activate PXR and target genes in the intestine much more robustly than in the liver. In acute and 30-day toxicology studies in mice, we did not find changes in lipid metabolism or associated pathology. In the kinase assay screen, a minimal effect of either FKK5 or FKK6 kinase inactivation or interference with their activation function (e.g., p300) was found. While we have only evaluated a host of enzymes and receptors, it is of interest that these indole analogs have limited cross-receptor activation as opposed to their parent indole metabolites from bacteria (Venkatesh *et al*, 2014). Thus, with this rationale, in the future, we will perform detailed head-to-head comparisons of our compounds against known PXR ligands for benchmarking efficacy and chronic toxicity.

In summary, the identified FKK lead compounds are suitable starting points for further optimization for preclinical drug discovery. Possible clinical applications comprise diseases in which increased intestinal permeability potentiates disease manifestations (e.g., inflammatory bowel disease) (Bischoff *et al*, 2014). Additionally, since we have demonstrated that these are relatively well-tolerated agents, applications in chemoprevention are also possible. Our work describes the first fully realized drug discovery using microbial metabolite mimicry. These strategies could be expanded widely to other receptors (and diseases) that derive ligands from the host microbiome.

likely that one explanation for a lack of lipid perturbation by FKK6 might be that PXR activation can have opposing effects on lipid metabolism depending on the nature of the xenobiotic itself (i.e., non-receptor-mediated action of the compound itself like antioxidant function) (Hwang *et al*, 2009). These effects may cancel out specific pathways like lipid metabolism (Hwang *et al*, 2009). We have studied a limited number of cellular targets that relate to PXR with respect to FKK6. While we find that FKK effects in cells and human intestinal organoids (HIOs) correlate with changes in PXR target genes and NF-κB signaling, it is possible that some of its protective effects could be related to other targets (e.g., tubulin) (La Regina *et al*, 2015). Loss of PXR studies in HIOs would be required to confirm these correlations; these studies are in progress using both LS174T cells and iPSC cells, in which PXR is knocked out using CRISPR/Cas9. We have shown FKK6 beneficial effect in the DSS-induced model of colitis. The use of mouse models beyond DSS-induced colitis (e.g., *IL-10*$^{-/-}$) could broaden the therapeutic

# Materials and Methods

## Chemistry, compound crystal structure, LEM analysis, and *In silico* experiments

See Appendix Supplementary Methods.

## Biology

Rifampicin, 2,3,7,8-tetrachlorodibenzo-*p*-dioxin (TCDD), DEX (dexamethasone), 1α,25-dihydroxyvitamin $D_3$, T3 (triiodo-L-thyronine), DHT (dihydrotestosterone), and SR12813 (S4194) were purchased from Sigma. LS174T cells were initially purchased from ATCC, and before use, cell line authenticity was validated as previously described (Wang *et al*, 2011). Caco-2 cells were purchased from ATCC, and authenticity was validated by multiplexed STR DNA (short tandem repeat deoxyribonucleic acid) profiling performed by Genetica. Human Caucasian colon adenocarcinoma cells LS180 (ECACC No. 87021202) were purchased from the European Collection of Cell Cultures (ECACC). Stably transfected gene reporter cell lines AZ-AHR, AZ-GR, IZ-VDRE [VDR elements (DNA binding)], PZ-TR, and AIZ-AR were as described elsewhere (Novotna *et al*, 2011, 2012; Bartonkova *et al*, 2015, 2016; Illes *et al*, 2015). Cells were cultured in Dulbecco's modified Eagle's medium (DMEM) or RPMI 1640 medium (AIZ-AR cells) supplemented with 10% fetal bovine serum, 100 U/ml streptomycin, 100 mg/ml penicillin, 4 mM L-glutamine, 1% non-essential amino acids, and 1 mM sodium pyruvate. Cells were maintained at 37°C and 5% $CO_2$ in a humidified incubator. HepaRG™ is a human hepatoma cell line isolated from a liver tumor of a female patient suffering from hepatocarcinoma and hepatitis C infection (Gripon *et al*, 2002). Three HepaRG™-derived cell lines, including 5F Clone control cells, and cells with targeted functional PXR and AhR gene knockouts, PXR-KO and AhR-KO, respectively, were purchased from Sigma (Czech Republic). Cells were handled according to the manufacturer's protocol. Human hepatocytes were purchased from Biopredic International (France) and Triangle Research Labs, LLC, Lonza (USA). Cultures were maintained in serum-free medium at 37°C and 5% $CO_2$ in a humidified incubator.

## Generation of PXR-knockout LS174T cell line

LS174T cells (ATCC: CL-188; 7 000 3535) (Synthego Corporation, Redwood City, CA) were the parental cells used to generate pooled clones of PXR-knockout. CRISPR/Cas9-mediated knockout cells were generated by Synthego Corporation (Redwood City, CA, USA). Briefly, cells were first tested negative for mycoplasma. Guide RNA was selected for high activity, specificity, and activity to create premature stop codons through frameshift mutations in the coding region via insertions and/or deletions (Indels) within exon 2 of the gene encoding human *PXR* (*NR1I2* transcript ID ENST00000337940). To generate specific guide RNAs, based on off-target analysis, the following modified RNAs were selected: *NR1I2-119,807,249 5′ AAGAGGCCCAGAAGCAAACC-3′ [TGG]-PAM*. To generate these cells, ribonucleoproteins (RNPs) containing the Cas9 protein and synthetic chemically modified sgRNA (Synthego) were electroporated into the cells using Synthego's optimized protocol. Editing efficiency is assessed upon recovery, 48 h post-electroporation.

Genomic DNA is extracted from a portion of the cells, PCR-amplified, and sequenced using Sanger sequencing. The resulting chromatograms are processed using Synthego Inference of CRISPR Edits software (ice.synthego.com). The pooled PXR-KO clones were validated via analysis of PXR protein expression and target gene induction activity through serial passages (Fig EV5A and B).

## Cell culture and transfection assays

LS180 cells transiently transfected by lipofection (FuGENE HD Transfection Reagent) with pSG5-PXR plasmid along with a luciferase reporter plasmid p3A4-luc (Goodwin *et al*, 1999; Huang *et al*, 2007) were used for assessment of PXR transcriptional activity. Transcriptional activity of AHR, GR, VDR, TR, and AR was studied in stably transfected gene reporter cell lines AZ-AHR, AZ-GR, IZ-VDRE, PZ-TR, and AIZ-AR, respectively (Novotna *et al*, 2011, 2012; Bartonkova *et al*, 2015, 2016; Illes *et al*, 2015). Cells were seeded in 96-well plates, stabilized for 24 h, and then incubated for 24 h with tested compound and/or vehicle DMSO (0.1% v/v) in the presence (antagonist mode) or absence (agonist mode) of rifampicin (RIF, 10 μM), 2,3,7,8-tetrachlorodibenzo-*p*-dioxin (TCDD; 5 nM), dexamethasone (DEX; 100 nM), 1α,25-VD3 (calcitriol, 50 nM), 3,5,3′-triiodothyronine (T3, 10 nM), or dihydrotestosterone (DHT, 100 nM), respectively. Incubations were performed in four technical replicates. After the treatments, cells were lysed and luciferase activity was measured on Tecan Infinite M200 Pro plate reader (Schoeller Instruments, Prague, Czech Republic). The data are expressed as fold induction ± SD of luciferase activity over the control cells (agonistic mode—in the absence of a model ligand) or as a percentage of maximal activation ± SD (antagonistic mode—in the presence of a model ligand). Differences were tested using one-way ANOVA with Dunnett's *post hoc* test, and $P < 0.05$ was considered significant (*). Transfections and analysis of AhR activation were performed as previously published (Murray *et al*, 2010). For all the transfection experiments, specifically multiplexed transfections, the control wells were mock-transfected (PXR vector). However, to exclude non-specific binding of PXR (or other factors) upstream of the luciferase gene within non-PXR-binding elements, we performed pilot experiments in different cell passages (P11, P14, and P17) of LS174T cells using a mock luciferase plasmid (luciferase not containing an upstream PXR-binding element of the CYP3A4 promoter). The PXR ligands tested were rifampicin (10 μM) and FKK6 (1–25 μM).

Reverse transfection assays were performed in Caco-2 cells as previously published with the following modifications (Kublbeck *et al*, 2015). $3.0 \times 10^4$ cells per well were transfected using pSG5-PXR plasmid (300 ng/well), β-Gal expression plasmid (600 ng/well), and CYP3A4-luciferase reporter (500 ng/well) (Huang *et al*, 2007). After 24-h transfection using Lipofectamine® LTX (1 μl/mg of plasmid), cells were exposed to FKK5 for 24 h. Cell lysates were prepared for β-Gal and luciferase assays as previously published (Huang *et al*, 2007). The data are expressed as mean ± SD RLU (relative light units from luciferase assay) normalized to β-Gal activity in the same well. Differences were tested using 2-way ANOVA with Tukey's multiple comparison test, and $P < 0.05$ was considered significant (*). Standard transient transfection assays (PXR, CAR) in HEK293T cells were performed as previously published (Huang *et al*, 2007).

**Table 2. Primer sets.**

| Gene symbol | Forward primer sequence | Reverse primer sequence |
|---|---|---|
| CYP3A4 | GGGAAGCAGAGACAGGCAAG | GAGCGTTTCATTCACACCACCA |
| MDR1 | AAAAAGATCAACTCGTAGGAGTA | GCACAAAATACACCAACAA |
| TLR4 | TGTGAAATCCAGACAATTGA | AAACTCTGGATGGGGTTTCCTG |
| GAPDH | TCTCTGCTCCTCCTGTTC | CTCCGACCTTTCACCCTTCC |

NF-κB reporter assays were performed using LS180 cells. Briefly, LS180 cells (passages 8 and 13) (Mastropietro *et al*, 2015) were transiently transfected using FuGENE HD Transfection Reagent with a reporter plasmid pNL3.2-NF-κB-RE[NlucP/NF-κB-RE/hygro] from Promega (Hercules, CA, USA), with or without co-transfection of the wt-PXR expression vector. Following 16 h of stabilization, the transfected cells were treated with a vehicle (DMSO, 0.1% v/v), rifampicin (RIF, 10 μM), or the tested compounds FKK5 or FKK6 in a concentration range from 0.1 to 25 μM for 24 h. For the last 4 h of the treatment, a combination of tested compounds or rifampicin with tumor necrosis factor α (TNF-α, 5 ng/ml) was applied. Thereafter, cells were lysed, and Nano luciferase activity was measured using a Tecan Infinite M200 Pro plate reader (Schoeller Instruments, Czech Republic).

### Quantitative real-time PCR

Cells or primary human hepatocytes from four different donors (HEP200529, HEP220932, HEP200533, and HEP200538) were incubated for 24 h with test compounds; the vehicle was DMSO (0.1% v/v). Total RNA was isolated using TRI Reagent® (Molecular Research Center, Ohio, USA). cDNA was synthesized from 1,000 ng of total RNA using M-MuLV Reverse Transcriptase (New England Biolabs, Ipswich, Massachusetts, USA) at 42°C for 60 min in the presence of random hexamers (New England Biolabs). Quantitative reverse transcriptase–polymerase chain reaction (qRT–PCR) was performed using LightCycler® 480 Probes Master on a LightCycler® 480 II apparatus (Roche Diagnostic Corporation). The levels of *CYP1A1, CYP1A2, CYP3A4, MDR1,* and *GAPDH* mRNAs were determined using Universal Probes Library (UPL; Roche Diagnostic Corporation) probes, and primers are listed in Table 1.

The following protocol was used: An activation step at 95°C for 10 min was followed by 45 cycles of PCR (denaturation at 95°C for 10 s; annealing with elongation at 60°C for 30 s). The measurements were performed in triplicate. Gene expression was normalized *per GAPDH* as a housekeeping gene. The data were processed according to the comparative $C_T$ method (Schmittgen & Livak, 2008). Data are expressed as fold induction ± SD over the vehicle-treated cells. The bar graph depicts one representative experiment of a series of experiments performed in three consecutive passages of cells. Differences were tested using one-way ANOVA with Dunnett's *post hoc* test, and $P < 0.05$ was considered significant.

In a separate set of experiments, $1 \times 10^5$ Caco-2 cells were transfected with pSG5-PXR for 12 h, then treated with the vehicle (0.1% DMSO), rifampicin, FKK5, and FKK6 (10 μM) in triplicate for 24 h. Cells were harvested for preparation of total RNA using TRIzol Reagent (Ambion, #15596026, Carlsbad, CA 92008) and

reverse-transcribed to cDNA using the High Capacity cDNA Reverse Transcription Kit (Thermo Fisher Scientific, #4368814, LT 02241). RT–qPCR was performed using Thermo Fisher PowerUp SYBR Green Master Mix (# A25742) and Thermo Fisher qPCR 7900HT. Each sample was repeated in quadruplicate and normalized to internal control, GAPDH. The entire experiment was repeated at least two separate times. The primer sequences are noted in Table 2.

### Cytotoxicity assays

For LS180 cells, the MTT colorimetric assay was performed as previously published (Schiller *et al*, 1992; Bartonkova *et al*, 2015) and using established proliferation assay kits (Vybrant MTT Cell Proliferation Assay Kit, Ref # V13154, lot # 1774057, Life Technologies).

### Chromatin immunoprecipitation (ChIP) assays

LS174T cells were transfected with pSG5-PXR (2 μg/plate/$1 \times 10^7$ cells). After 12 h, cells were exposed to vehicle (DMSO 0.1% v/v) or 10 μM FKK5, FKK6, or FKK9 for 24 h. Chromatin immunoprecipitation (ChIP) was performed using previously published protocols (Nelson *et al*, 2006; Wang *et al*, 2011). Briefly, cells were chemically cross-linked with formaldehyde in culture media for 15 min at room temperature. Glycine was added to a final concentration of 0.125 M to stop cross-linking for 5 min at room temperature. Cells were rinsed twice with cold PBS (phosphate-buffered saline) (pH 7.5) and scraped in PBS supplemented with PIC (protease inhibitor cocktail).

Cell pellets were collected and immediately resuspended in immunoprecipitation (IP) buffer (150 mM NaCl (sodium chloride), 50 mM Tris–HCl at pH 7.5, 5 mM EDTA (ethylenediaminetetraacetic acid), 0.5% NP-40, 1% Triton X-100). The nuclei were centrifuged at $12,000 \times g$ for 1 min and resuspended in fresh IP buffer. Chromatin was sheared by sonication to an average size of 0.2-0.9 kb. Lysates were cleared by centrifugation at $12,000 \times g$ for 10 min. Chromatin from $2 \times 10^6$ cells (about 200 μl) was incubated overnight at 4°C with the following 2 μg rabbit anti-PXR [sc-25381 (H-160), Santa Cruz Biotechnology, CA] or normal rabbit IgG (sc-2027, Santa Cruz Biotechnology, CA). Immune complexes were captured with 20 μl of packed Protein A Agarose (sc-2001, Santa Cruz Biotechnology) per IP for 1 h at 4°C and washed three times with IP buffer. 10% (wt/vol) Chelex 100 was added to the beads followed by boiling of the suspension and centrifugation at $12,000 \times g$ for 1 min to obtain chromatin.

Purified chromatin DNA was used to perform RT–qPCR. We used PowerUp SYBR Green Master Mix (A25742, Thermo Fisher Scientific, TX) in a 10 μl reaction, 0.5 μl DNA template, 0.5 μl primer pairs (10 μM each), 5 μl Master Mix, and 3.5 μl $H_2O$ in 384-well plates on an ABI 7900 (default three-step method, 40 cycles). Data were analyzed using SDS 2.2.1 program (ABI Biotechnology). The primers used for ChIP are shown in Table 3.

### Expression and purification of His-tagged PXR ligand-binding domain (LBD) protein

The cloning, expression, and purification of His-tagged PXR-LBD were performed as previously published (Wallace *et al*, 2013) with

**Table 3. Primer sets.**

| Gene symbol | Forward primer sequence | Reverse primer sequence |
| --- | --- | --- |
| CYP3A4 | ATGCCAATGGCTCCACTTGAG | CTGGAGCTGCAGCCAGTAGCAG |
| MDR1 | ACCAACTGTTCATTGGTCTGC | GCAATCAGCTTAGTACCTGGATG |
| CYP1A1 | AGCTAGGCCATGCCAAAT | AAGGGTCTAGGTCTGCGTGT |

the following modifications: For protein expression, Luria–Bertani (LB) media were inoculated with a saturated culture of BL21-Gold cells transformed with HIS-LIC plasmid containing the PXR-LBD construct. The mixture was allowed to shake at 37°C until the cells reached an $OD_{600} \sim 0.6$, and then, the temperature was reduced to 18°C, at which time IPTG was added (final concentration of 0.1 mM) to induce protein expression. For purification, the His-tag was not removed, and the uncleaved protein was loaded onto the gel filtration column with buffer containing HEPES (4-(2-hydroxyethyl)-1-piperazineethanesulfonic acid) (25 mM, pH 7.5) and NaCl (150 mM).

## Isothermal titration calorimetry (ITC)

Isothermal titration calorimetry (ITC) was carried out using a VP-ITC microcalorimeter from MicroCal/Malvern Instruments (Northampton, MA, USA). The protein and its ligands were prepared in 25 mM HEPES, pH 7.5, with 150 mM NaCl and DMSO at a concentration of 6% for the experiment with FKK5 and FKK6 and 2% in the trial with rifampicin and 3-IPA. In all the experiments, the ligand solution was injected in 10 µl aliquots into the calorimetric cell containing PXR-LBD (ligand-binding domain) at a concentration of 3–6 µM. The respective concentrations of FKK5, FKK6, rifampicin, and 3-IPA in the syringe were 60, 80, 330, and 400 µM. The experiments were carried out at 37°C. The heat evolved upon each injection of the ligands was obtained from the integral of the calorimetric signal. The heat associated with binding to PXR-LBD in the cell was obtained by subtracting the heat of dilution from the heat of reaction. The individual heats were plotted against the molar ratio, and the enthalpy change ($\Delta H$), association constant ($K_a = 1/K_d$), and the stoichiometry were obtained by non-linear regression of the data. In the case of rifampicin, a binding model was chosen that took into account two sets of sites with different binding affinities.

## hPXR TR-FRET

The PXR ligand-binding assay of FKK5, FKK6, and FKK9 was performed using LanthaScreen TR-FRET PXR (SXR) Competitive Binding Assay Kit (PV4839; Invitrogen, USA) according to the manufacturer's instructions. The assays were done with concentrations of tested compounds ranging from 1 nM to 25 µM. DMSO and 100 µM SR12813 were used as a negative and positive control, respectively. The reaction mixture was incubated at room temperature for 1 h in the dark, and then, fluorescent signals were measured at 495 and 520 nm, with the 340-nm excitation filter, on Infinite F200 microplate reader (Tecan Group Ltd, Switzerland). Finally, the TR/FRET ratio was calculated by dividing the emission signal at 520 nm by that at 495 nm. All PXR-binding assays were performed as two independent experiments, each with a minimum of four replicates. Final $IC_{50}$ was obtained by processing the data with GraphPad Prism 6 using standard curve interpolation (sigmoidal, 4PL, variable slope). SR12813 served as a positive control PXR agonist ligand, and it demonstrates an $IC_{50}$ of 0.127 µM, which is similar to previously published results (Shukla *et al*, 2009).

## Human studies

Human tissues (duodenum) were collected according to the standard research protocols approved by the Institutional Review Board and Department of Pathology at Cincinnati Children's Hospital (IRB: 2014-6279; renewed 11/27/2017).

Additional samples were collected from consenting healthy patients undergoing colonoscopy at the University of Calgary endoscopy unit for colon cancer screening or to investigate gastrointestinal symptoms with normal colonoscopic appearance and normal histology (Study ID: REB18-0631_REN1). Biopsies via endoscope were taken from the colonic mucosa and immediately placed in Intesticult^tm Organoid Growth Media supplemented with antibiotic/antimycotic (Stemcell Technologies) and transferred from the unit to the laboratory.

In both the Cincinnati and Calgary study, informed consent was obtained from all subjects, and the experiments conformed to the principles set out in the WMA Declaration of Helsinki and the Department of Health and Human Services Belmont Report.

## Isolation of crypts and culture of human enteroids from patient-derived duodenum (and RT–PCR assays)

Discarded duodenum tissue after surgery was obtained and used to isolate crypts. Embedded crypts in Matrigel formed 3D structures referred to as "enteroids". The isolation protocol has been described (Mahe *et al*, 2015). Enteroids were incubated with 10 µM compounds (FKK5, FKK6, and rifampicin) at day 4 from isolation. DMSO was added to enteroids as a negative control. After incubation for 24 h, enteroids were collected in RNA-free tube followed by breaking down Matrigel by pipetting vigorously (using 1 ml PBS). The enteroids were pelleted at 16,000 *g* for 3 min (4°C). RNA was extracted using miRNA isolation kit (Invitrogen; #AM1561) using the protocol provided by Invitrogen. The cDNA was synthesized from the extracted RNA (final amount of RNA: 1 µg) using SuperScript III (Invitrogen, #18080-051) through two-step procedure provided by Invitrogen. PowerUp SYBR Green (Applied Biosystem, #A25742) was used for RT–PCR and performed 40 cycles using Quant Studio3 (Invitrogen, #A28132). The primer sets are listed in Table 4.

## Culture of human-derived colonic enteroids and FKK treatment assays (Calgary Protocol)

Samples were immediately processed to yield isolated colonic crypts as previously described (Fernando *et al*, 2017). Isolated colonic crypts were embedded in Matrigel® (Corning) and cultured in Intesticult^tm Organoid Growth Media. Human organoids were pretreated for 6 h with either FKK5, FKK6, FKK9, or vehicle (DMSO), then stimulated with human recombinant TNF-alpha in the presence of either treatment for an additional 12 h. Organoids were washed in

**Table 4. Primer sets.**

| Gene symbol | Forward primer sequence | Reverse primer sequence |
|---|---|---|
| CYP3A4 | AGATGCCTTTAGGTCCAATGGG | GCTGGAGATAGCAATGTTCGT |
| MDR1 | TTGCTGCTTACATTCAGGTTTCA | AGCCTATCTCCTGTCGCATTA |
| ABCC2 | CCCTGCTGTTCGATATACCAATC | TCGAGAGAATCCAGAATAGGGAC |
| UGT1A1 | CATGCTGGGAAGATACTGTTGAT | GCCCGAGACTAACAAAAGACTCT |
| GSTA1 | CTGCCCGTATGTCCACCTG | AGCTCCTCGACGTAGTAGAGA |

PBS, disrupted in Tri Reagent® (Sigma, Oakville, ON, CAN), and frozen at −80°C. After thawing, chloroform was added, samples were centrifuged for 15 min, and the resulting aqueous phase was then added to equal volumes of 70% ethanol and further processed using the RNeasy mini kit (Qiagen). cDNA was synthesized using the QuantiTect RT kit (Qiagen) according to the manufacturer's protocol. Resulting cDNA was used as a template for quantitative real-time PCR using Perfecta SYBR Green FastMix with ROX (QuantaBio). PCR and analysis were performed using a StepOne PCR System (Applied Biosystems). Gene expression was calculated relative to β-actin expression and expressed as fold change of control. The primers used were as follows: Human®-Actin (NM_001101), Human CYP3A4 (NM_001202855), and Human ABCB1 (NM_000927).

### Caco-2 cell cultures and iPSC-derived human intestinal organoids (and RT–PCR assays)

Caco-2 cell (ATCC, Manassas, VA) passages 25–35 were maintained and expanded in culture flasks in Dulbecco's modified Eagle's medium (DMEM) with 10% FBS and 1× antibiotic–antimycotic. The cells were kept in a humidified 37°C incubator with 5% $CO_2$. Media were changed every 2–3 days, and cells were passaged 1–2 times a week. For all experiments, Caco-2 cells ($5 \times 10^4$ cells/ml) were seeded into 12-well Transwell inserts or coverslips and cultured for 3 weeks. HIOs were generated from iPSCs as described previously (Spence et al, 2011), with some modifications. The iPSC cell line IISH1i-BM1 (WiCell, Madison, WI) was cultured on Matrigel plates to 80% confluence in mTeSR™ 1 medium (Stem Cell Technologies, Seattle, WA) and differentiated to endoderm using Definitive Endoderm Kit (Stem Cell Technologies., Seattle, WA) for 4 days with daily media changes. Endoderm was then differentiated to hindgut for 5 days with the addition of FGF-4 (500 ng/ml; Peprotech, Rocky Hill, NJ) and Chir99021 (3 μM; Cayman, Ann Arbor, MI) in RPMI media supplemented with 2% FBS. Floating hindgut spheroids were collected on days 3–5 of hindgut differentiation and suspended in Matrigel beads for HIO differentiation. HIOs were maintained in WENRAS media (Fujii et al, 2015), which maintains proliferation in the culture. Stock HIOs were passaged weekly by removing Matrigel with Cell Recovery Solution (Corning, NY) and mechanically breaking up organoids with pipetting. HIOs were seeded into new Matrigel beads at a 1:3 split, using WENRAS with added Rock Inhibitor Y-27632 (10 μM) and Chir99021 (3 μM; Cayman., Ann Arbor, MI) for the first 2 days of culture.

HIOs and Caco-2 were exposed to a pro-inflammatory cytokine cocktail (10 ng/ml IFN-γ, 10 ng/ml TNF-α, 50 ng/ml IL-1β) for 2–24 h in combination with DMSO or FKK5 and FKK6 compounds (10 and 25 μm). Relative mRNA levels of IL-8 and IL-6 were determined using quantitative real-time PCR, with normalization to GAPDH. RNA was isolated using an RNeasy Mini Kit (Qiagen) as per the manufacturer's instructions. On column DNase (Qiagen) was used to remove any contaminating DNA. Next, cDNA was formed using iScript cDNA Synthesis Kit (Bio-Rad) and 100 ng RNA. qRT–PCR was performed with SsoAdvanced Universal SYBR Green Kit (Bio-Rad). The following primers were used: IL-8 Forward: ATACTCCAAACCTTTCCACCC; IL-8 Reverse: TCTGCACC CAGTTTTCCTTG; IL-6 Forward: CCACTCACCTCTTCAGAACG; IL-6 Reverse: CATCTTTGGAAGGTTCAGGTTG; GAPDH Forward: AC ATCGCTCAGACACCAT; GAPDH Reverse: TGTAGTTGAGGTCAA TGAAGGG. For immunofluorescent staining of NF-κB translocation to the cell nucleus, HIOs and Caco-2 cell monolayers were fixed in 4% formaldehyde (in 1 × PBS) overnight at 4°C. The formaldehyde was removed by washing with 1 × PBS, and an antigen retrieval step was performed by incubating in 10 mM sodium citrate buffer in a vegetable steamer for 20 min. The samples were then blocked with normal donkey serum (10% in PBS with 0.3% Triton X-100) for 1 h. Samples were incubated overnight with primary antibodies: rabbit anti-NF-κB p65 (D14E12 # 8242, 1:400 dilution, Cell Signaling Technology, Danvers MA), then immersed in Alexa Fluor® 555 donkey anti-rabbit secondary antibody (A-31572, Life Technologies) at a 1:500 dilution. Nuclei were stained with Hoechst 33342 (Life Technologies). The samples were then imaged using a Zeiss LSM880 Confocal/Multiphoton Upright Microscope, with 3-D image rendering using Volocity. For controls, normal rabbit IgG (#2729S, Life Technologies) was used.

FKK compounds were tested for their ability to prevent Salmonella invasion in vitro in non-inflamed and inflamed cell cultures. Overnight cultures of S. typhimurium 14038 were diluted to $OD_{600}$ 0.01 and injected into with HIOs or immersed over Caco-2 monolayers for 1 h at 37°C. The invasive ability of Salmonella was assessed using the gentamicin protection assay (Gagnon et al, 2013). Bacterial adhesion scenarios to the Caco-2 surface were set up as described above. Non-adhered bacteria were removed by washing twice in PBS, followed by incubation with 1 ml gentamicin (150 μg/ml in DMEM) for 1 h at 37°C to kill the adhered extracellular bacteria. Dead bacteria were removed by washing twice in PBS, followed by incubation with 500 μl 0.1% Triton X-100 for 15 min at 37°C to lyse the HIOs and Caco-2, and release the intracellular (invaded) bacteria. Serial fold dilutions and plating were then employed to determine CFU/ml. We normalized to total protein using Bradford assay.

### Animal studies

FKK6 was solubilized to saturation concentrations (~ 500 μM) at room temperature in 10% DMSO (DMSO maximum tolerated dose in mice is ~ 2.5 g/kg/day for 35 consecutive days) (Caujolle et al, 1967a; Brayton, 1986). Since we do not know the bioavailability of FKK6 in mice, a conservative guide is that 10-90% of the administered dose for any given mouse is absorbed. This approach would ensure the most mice would exhibit concentrations exceeding

50 μM in the feces, which would be higher than 10 μM concentrations used in cell culture.

Male and female C57BL/6J mice (Jackson Labs Stock 000664) (6–10 weeks), $pxr^{-/-}$ (6–10 weeks) (Staudinger *et al*, 2001), and male and female mice expressing the human PXR gene (*hPXR*) [Taconic, C57BL/6-*Nr1i2*$^{tm1(NR1I2)Arte}$ (9,104, females; 9,104, males), 6–10 weeks] (Scheer *et al*, 2010) were administered 10% DMSO ($n = 3$) or FKK6 ($n = 3$) in 3 or 5 doses, with each dose interval of 12 h. Each gavage dose volume was 100 μl. 2–4 h after the dosing period, mice were sacrificed, and tissues were harvested for further analysis. The mouse studies were performed three independent times over an 18-month period. All purchased mice were acclimatized in the Einstein vivarium for ~ 2 weeks before experiments started. All mice ($n \leq 5$ per stainless steel cage and separated by gender) were maintained on standard non-irradiated chow (LabDiet 5058, Lab Supply) and sterile water in the barrier facility observing a 12-h night/day cycle. The room temperature was maintained at $25 \pm 2°C$ with $55 \pm 5\%$ humidity. All studies were approved by the Institute of Animal Studies at the Albert Einstein College of Medicine, INC (IACUC # 20160706, 20160701 and preceding protocols), and specific animal protocols were also approved by additional protocols (IACUC# 20170406, 20170504) and Animal Care and Use Review Office (ACURO) of the US Army Medical Research and Materiel Command (PR160167, W81XWH-17-1-0479).

**Dextran sulfate sodium (DSS)-induced acute colitis studies**

For these studies, 7- to 8-week-old female (DeVoss & Diehl, 2014) *hPXR* (mice expressing the human PXR gene) and $pxr^{-/-}$ mice (C57BL/6 background) were bred and each genotype was separated into two treatment groups using a random allocation method via coin toss—(i) $n = 3$ control (0.8% DMSO in 100 μl drinking water) and (ii) $n = 3$ FKK6 (200 μM (0.4 mg/kg/day) in 0.8% DMSO in 100 μl drinking water) administered once per day via oral gavage, and via intrarectal gavage for a total of 10 consecutive days The dose was chosen based on the extrapolation of the total dose delivered over 10 days to mice (~ 4 mg/kg), which is equivalent to that delivered to mice for determining target gene expression over 3–5 doses (3–5 mg/kg). Intrarectal delivery, at a 45° angle, was achieved using a 1-ml syringe attached to a 19G needle that had a polyethylene tubing (#427516; 0.58 mm O.D, Becton Dickinson, Sparks, MD) inserted to 3 cm into anal opening under very light anesthesia (1.5% isoflurane with nose cone) till no visible liquid was seen outside the anus. Fecal pellets were collected prior to the initiation of the treatment protocol. All mice were starved overnight (12 h) to allow for smooth intrarectal delivery; however, the mice were allowed to feed *ad libitum* during the day. After 10 days, the mice were sacrificed and organs/tissues/feces harvested for further study. Prior to necropsy, mice were administered FITC-dextran, and mouse serum was collected for the FITC-dextran assay as previously described (Vetizou *et al*, 2015). The experimenter was not blinded to treatment allocation; however, the pathologist (KS) evaluating histologic scores was blinded to treatment allocation. All studies were approved by the Institute of Animal Studies at the Albert Einstein College of Medicine, INC (IACUC # 20160706 and preceding

**Table 5. Primer sets.**

| Gene symbol | Forward primer sequence | Reverse primer sequence |
|---|---|---|
| TBP | ACCGTGAATCTTGGCTGTAAAC | GCAGCAAATCGCTTGGGATTA |
| GAPDH | AACTTTGGCATTGTGGAAGG | GGATGCAGGGATGATGTTCT |
| IL-10 | TGAGGCGCTGTCGTCATCGA TTTCTCCC | ACCTGCTCCACTGCCTTGCT |
| TNF-R2 | CAGGTTGTCTTGACACCCTAC | CACAGCACATCTGAGCCT |
| TNF-α | ATGAGAAGTTCCCAAATGGC | AGCTGCTCCTCCACTTGGTGG |
| IL-17f | TGCTACTGTTGATGTTGGGAC | AATGCCCTGGTTTTGGTTGAA |

protocols). Colitis scoring was performed as previously published, and post-examination masking was conducted (Kim *et al*, 2012; Meyerholz & Beck, 2018).

For RT–qPCR of colon tissue for cytokine expression data, new experiments were conducted with 3% DSS and the identical FKK6 dose ($n = 3$ mice/group) and scheduled as above. RT–qPCR was performed using PowerUp™ SYBR® Green Master Mix (A25741, Thermo Fisher Scientific) and primers specific for IL-10, TNF-α, TNFR2, and IL-7f. Each target gene was analyzed in quadruplicate. Normalizing controls included GAPDH (used to illustrate data) and TBP. The primer sequences used are shown in Table 5.

**Statistical analysis**

All data sets were assessed for normality and lognormality, descriptive, and outlier analyses using tests specified in GraphPad Prism 8.2.0 (272) (except as noted). Student's $t$-test, one-way analysis of variance (1-way ANOVA), two-way analysis of variance (2-way ANOVA) followed by Dunnett's test, and values of $EC_{50}$ and $IC_{50}$ calculated using GraphPad Prism, version 8.2.0 (272) for Windows (GraphPad Software, La Jolla, California, USA), are indicated as appropriate in each figure legend. The statistical analysis of FKK6 treatment endpoints (% weight loss, colon length, lipocalin 2, and FITC-dextran) in $pxr^{+/+}$ and $pxr^{-/-}$ was performed using Student's $t$-test and non-parametric tests for non-normal data and by clustering method on the six sample data vector to split the mice into two groups. The "kmeans" function of the R software was used to obtain the clusters.

## Data availability

The FKK5-PXR docking model has been submitted to BioModels with identifier: MODEL2002050001 (https://www.ebi.ac.uk/biomodels/MODEL2002050001)[#]. The immunofluorescence images have been submitted to BioStudies with identifier: S-BSST310 (https://www.ebi.ac.uk/biostudies/preview/studies/S-BSST310). FKK5 and FKK6 structures have been deposited in the Cambridge Structural Database under deposition numbers 1948848 and 1948849, respectively.

**Expanded View** for this article is available online.

[#]Correction added on 7 May 2020, after first online publication: the BioModels accession ID has been updated.

## The paper explained

### Problem

The traditional approaches to drug discovery (e.g., small molecule library screens) are limited by the unpredictable toxicity of the potentially effective new drugs. Indeed, virtually any aspect of human health is dependent on a pipeline of drugs that may or may not enter the clinic due to overt toxicities observed during its development in the laboratory and clinic. To overcome this problem, the concept of expanding the chemical space within drug discovery on the basis of microbial metabolite mimicry has been proposed. These mimics are non-toxic but potent modulators of the target and disease.

### Results

We have previously defined weak agonists of the human pregnane X receptor (PXR, a xenobiotic nuclear receptor) that involved co-activation via the microbial L-tryptophan metabolites, indole and indole 3-propionic acid. Based on the pharmacophores of the binding of these metabolites with the PXR ligand-binding domain, two lead analogs, FKK5 and FKK6, were developed based on favorable PXR interaction profiles in cells. Unlike indole, which activates the aryl hydrocarbon receptor (AhR), FKK5 and FKK6 did not activate AhR and several other related xenobiotic receptors. In general, after exposure to FKK5 or FKK6, intestinal cell lines (LS180, LS174T) showed a more robust expression of PXR target genes than hepatic cell lines. FKK5 and FKK6 showed direct interaction with PXR with an affinity in the low micromolar range. FKK6 also enhanced both CYP3A4 and MDR1 promoter occupancy in cells. In an intestinal cell line model (LS180 and LS174T cells), FKK5 and FKK6 inhibited NF-κB activation in a PXR-dependent manner. In Caco-2 cells and iPSC-derived human intestinal organoids, FKK5 and FKK6 inhibited cytokine and salmonella infection-induced NF-κB nuclear translocation. In acute (10-day) and chronic (30-day) toxicity studies in mice, FKK6 does not induce any clinical toxicity. FKK6 induced PXR target gene cyp3a11 in mice expressing the human PXR transgene, but not in conventional wild-type mice or $pxr^{-/-}$ mice. Finally, FKK6 abrogated dextran sulfate sodium-induced colitis in mice expressing the human PXR gene. Together, these results suggest that the concept of microbial metabolite mimicry could indeed yield potential non-toxic drug leads.

### Impact

Our work describes the first fully realized drug discovery using microbial metabolite mimicry. The applications would be in fields in which deficiency in intestinal permeability alters (worsens) disease manifestations (e.g., inflammatory bowel disease). Additionally, since these are relatively well-tolerated agents, applications in chemoprevention are also possible. These strategies could be expanded widely to other receptors (and diseases) that derive ligands from the host microbiome.

## Acknowledgements

The studies presented here were funded in part by the ICTR Pilot Award (AECOM to S.M & F.K); Grant# 362520 Broad Medical Research Program (BMRP, not Litwin) at CCFA (Crohn's & Colitis Foundation of America) (to S.M); NIH grants R35 ES028244 (to G.H.P); CA127231, CA 161879, and Department of Defense Partnering PI (W81XWH-17-1-0479; PR160167) (to S.M.); ES030197 (to S.M., J.C., H.G), as well as R43DK105694 (PI: Jay Wrobel) and P30DK041296 (PI: Alan Wolkoff) (Pilot Awards, S.M); Diabetes Research Center Grant (P30 DK020541); Cancer Center Grant (P30CA013330 PI: David Goldman); 1S10OD019961-01 NIH Instrument Award (PI: John Condeelis); LTQ Orbitrap Velos Mass Spectrometer System (1S10RR029398); and NIH CTSA (1 UL1 TR001073). Additional invaluable assistance was obtained from Vera DesMarais PhD (Light Microscopy and Image Analysis Analytical Imaging Facility (AIF), Albert Einstein College of Medicine, Bronx, NY), Amanda Beck DVM (Histology and Comparative Pathology Core, Albert Einstein College of Medicine, Bronx, NY), Lars Nordstroem PhD and Chamini Karunaratne PhD (Chemical Synthesis and Biology Core, Albert Einstein College of Medicine, Bronx, NY), Yungping Qiu PhD (Stable Isotopes and Metabolomics Core Facility, Albert Einstein College of Medicine, Bronx, NY, and the Proteomics Core Facility, Albert Einstein College of Medicine, Bronx, NY), NIH R01 CA207416 (to M.R.R.) and the Czech Science Foundation [19-00236S] and the Operational Programme Research, Development and Education—European Regional Development Fund, the Ministry of Education, Youth and Sports of the Czech Republic [CZ.02.1.01/0.0/0.0/16_019/0000754] (Z.D.), and Christian Jobin (E. coli NC101 strains, University of Florida, Gainesville, FL). Arpan De performed the biofilm assays. MMC and GL are indebted to professor Cele Abad-Zapatero who invented AtlasCBS and disclosed its potential as a tool to facilitate drug design and development endeavors. SV and AC are supported by the National Academy of Sciences, India. The authors thank Ms. Gurmeet Bindra in the inflammatory intestinal tissue bank (IITB) and the staff at the University of Calgary endoscopy unit for assistance with research sample collection and Dr. Marilyn Gordon of the Human Tissue Research Lab at the University of Calgary for sample preparation and human intestinal organoid protocol optimization. SAH's salary is supported by the CIHR's Canada Research Chair program (Tier II CRC in Host-Microbe Interactions and Chronic Disease); SAH's lab is supported operating funds from Crohn's & Colitis Canada; the Dr. Lloyd Sutherland Investigatorship in IBD/GI Research; Natural Sciences and Engineering Research Council (NSERC) Discovery Grant (RGPIN-2016-03842). Weijie Chen developed the final corrected figures. Arne Schön was supported by NIH funded contract (HHSN261200800001E).

## Author contributions

All authors contributed to the writing of the manuscript and have given approval to the final version of the manuscript. ZD performed conceptual and experimental design, data interpretation of TR/FRET, gene expression in cell lines, HepaRG, and human hepatocytes, and reporter gene assays (PXR, AHR, VDR, TR, GR, AR, NFκB); provided materials; and helped in writing the manuscript. FK originated and synthesized FKK1-10 chemicals. CMC and JSK carried out Caco-2 and HIO experiments and salmonella invasion studies. HL carried out all mouse studies, select transfection studies, and ChIP assay. AV contributed to the experimental assays, data acquisition, data evaluation of reporter gene assays (PXR, AHR), and gene expression in intestinal cell lines (transfections, treatments, RT–PCR) and in human hepatocyte cultures Hep200529 + Hep220932. MŠ contributed to the experimental assay, data acquisition, and data evaluation of gene expression in HepaRG and in human hepatocyte cultures Hep200533 + Hep200538 (RT–PCR). IB contributed to the experimental assay, data acquisition, and data evaluation of reporter gene assays (PXR, AHR, VDR, TR, AR, GR, NFκB); performed HepaRG (WT, PXR-KO, AHR-KO) cultivation, differentiation, RT-PXR analyses, and Western blots for validation of receptor expression; and contributed to the validation of LS174 WT/PXR-KO cells using RT–PCR and human hepatocyte cultures Hep200533 + Hep200538 (RT–PCR). EJ contributed to the general experimental analysis, data acquisition, and data evaluation of TR-FRET/PXR. KP and BV performed experimental analysis, data acquisition, data evaluation of reporter gene assays in intestinal cells (NFκB, co-transfected PXR), and mock experiments for PXR and AHR protein levels by Sally Sue. LUN performed and developed crystals of FKK5 and FKK6 in solution. CK scaled up synthesis of some FKK compounds. HR conducted reverse transfection studies with PXR. KSM generated organoid data in Cincinnati. APN contributed to conceptual and experimental design of organoid data generated in Cincinnati. IM and GHP externally validated effects of FKK compounds on AhR reporter assays. JB and LT contributed to the experimental design, data acquisition, and data

evaluation of the radioligand-binding assay for RXR. AS performed and analyzed isothermic titration calorimetry/ITC studies. WGW and MRR generated soluble PXR protein for ITC. KS and AB performed blinded pathology readings. SK performed PXR docking studies and contributed to the design of indole analogs. MCN performed X-ray diffraction analysis of FKK compounds in the solid state. S. Vishveshwara and A. Chandran performed PXR docking studies. MMC and GL performed LEM analysis. JYC and HG developed extraction methods for fecal analysis of FKK5 and FKK6 and MS verification of compound. JCM performed Caco-2 and HIO experimental design and setup, and salmonella invasion studies. SC carried out statistical analysis for all studies performed. AM and DW provided expert advice on animal models pertinent to barrier dysfunction and on FKK compound synthesis. KLF and SAH performed organoid studies in Calgary. RBS helped with identification of cytokines for study and interpretation of DSS model. SM contributed to initial conceptual design of microbial metabolite mimicry, planned all the studies presented in the paper, and developed the team and troubleshooting of all experiments presented.

## Conflict of interest

The studies presented here are included in a patent submitted by the Albert Einstein College of Medicine in conjunction with Palacký University and the Drexel University College of Medicine to the US Patent and Trademark Office. Funding for these studies is listed under acknowledgment.

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
