## [Review Process File · EMBO Molecular Medicine]

Targeting the Pregnane X Receptor Using Microbial Metabolite Mimicry

Zdeněk Dvořák, Felix Kopp, Cait M. Costello, Jazmin S. Kemp, Hao Li, Aneta Vrzalová, Martina Štěpánková, Iveta Bartoňková, Eva Jiskrová, Karolína Poulíková, Barbora Vyhlídalová, Lars U. Nordstroem, Chamini Karunaratne, Harmit Ranhotra, Kyu Shik Mun, Anjaparavanda P. Naren, Iain Murray, Gary H. Perdew, Julius Brtko, Lucia Toporova, Arne Schon, William G. Wallace, William G. Walton, Matthew R. Redinbo, Katherine Sun, Amanda Beck, Sandhya Kortagere, Michelle C. Neary, Aneesh Chandran, Saraswathi Vishveshwara, Maria M. Cavalluzzi, Giovanni Lentini, Julia Yue Cui, Haiwei Gu, John C. March, Shirshendu Chatterjee, Adam Matson, Dennis Wright, Kyle L. Flannigan, Simon A. Hirota, R. Balfour Sartor, Sridhar Mani

Review timeline:

Submission to The EMBO Journal	11th Oct 2019
Editorial decision	16th Oct 2019
Transfer to EMBO Molecular Medicine	16th Oct 2019
Editorial Decision:	5th Nov 2019
Revision received:	8th Jan 2020
Editorial Decision:	30th Jan 2020
Revision received:	4th Feb 2020
Accepted:	7th Feb 2020

Editor: Lise Roth

Transaction Report:

Editorial Decision from The EMBO Journal

16 Oct 2019

Thank you for submitting your manuscript "Targeting the Pregnane X Receptor Using Microbial Metabolite Mimicry" to The EMBO Journal. I have now read your study carefully and discussed it with my colleagues. I regret to inform you that we have decided not to pursue the publication at The EMBO Journal, but I recommend a transfer to our sister journal EMBO Molecular Medicine, where the responsible editor would be interested in sending the manuscript for peer review.

We appreciate that your study describes derivation of novel indole compounds for activation of PXR in treatment of intestinal inflammation. Two of the synthesized compounds activate PXR and alleviate intestinal and cellular inflammation without observable organ toxicity even after prolonged treatment. While we appreciate the medical relevance of the study, I am afraid that, since the more general function of indole metabolites in activation of PXR signalling in the context of intestinal inflammation was demonstrated in earlier work from your lab, the broader novelty of the provided insight into the molecular mechanisms of indole compound action in inflammation is not sufficient to consider publication at The EMBO Journal.

That being said, I appreciate the medical relevance of your study and have discussed the manuscript with my colleague Lise Roth at our sister journal EMBO Molecular Medicine, and she would be interested in sending your manuscript for external peer review.

Thank you for the submission of your manuscript to EMBO Molecular Medicine. We have now received feedback from the three reviewers who agreed to evaluate your manuscript. As you will see from the reports below, the referees acknowledge the interest of the study and are overall supporting publication of your work pending appropriate revisions.

Addressing the reviewers' concerns in full will be necessary for further considering the manuscript in our journal, and acceptance of the manuscript will entail a second round of review. EMBO Molecular Medicine encourages a single round of revision only and therefore, acceptance or rejection of the manuscript will depend on the completeness of your responses included in the next, final version of the manuscript. For this reason, and to save you from any frustrations in the end, I would strongly advise against returning an incomplete revision.

***** Reviewer's comments *****

Referee #1 (Comments on Novelty/Model System for Author):

I have few criticisms - this is a step forward for the field (and for all interested in microbial metabolomics in disease therapeutics), makes its points with clear experimental data and is clearly presented.

Referee #1 (Remarks for Author):

I enjoyed the experimental approach, the novelty, the choice of experimental models, the comprehensiveness of the study.

It seems churlish in this context perhaps to make any further comments or requests, but...

1. Some will find the generic references to 'humanized mice' in the DSS study confusing - it took me a while to spot this meant hPXR mice and not the more common use of the term, human immune system reconstituted mice such as Hu-SCID.
2. Figure 5 is quite an important selling point in the paper, but it seems a shame to have done these experiments without characterizing the nature of protection more fully in terms of transcriptional changes and, especially, looking more widely at cytokine changes. For example, any differences in IL-17 and IL-10?

Referee #2 (Comments on Novelty/Model System for Author):

Identify potent and selective PXR activators could potentially benefit the treatment of a number of diseases including intestinal inflammation.

Referee #2 (Remarks for Author):

In this manuscript, the authors chemically modified a number of microbial metabolites to generate highly selective and potent PXR activators. Given the promiscuous target gene spectrum of PXR and its cross-talk with many other nuclear receptors, selective PXR agonists are highly demanded for defining the exact functional nature of this receptor. Using microbial metabolite mimicry, the authors systematically evaluate two candidate PXR activators from a class of indole-derivatives. In addition to testing prototypical PXR activity, target gene expression, and promoter bindings, this manuscript also characterized the role of FKK6 (one of the indole-derivatives) in repressing pro-inflammatory cytokine production in mice. Overall, the experiments were well-designed and data were reasonably interpreted. The manuscript provides novel information regarding the potential clinical use of a class of PXR activators with low toxicity.

Specific comments:

1. The order of some figures and their panels should be reorganized. Often the result section already described part of figure 2 and 3 then started to talk figure 1d and 1e. This confusion also appeared in other figures.
2. Figure 1d bottom, the quantitative PCR data should be provided with mean values and standard deviations.
3. Figure 3a and 3d, it's unclear what the * indicates for; which groups were compared. It would be helpful if arrows or other indicators were placed in fig 3c and 3f to clarify the localization of NF- κ B.
4. Figure 4d showed very large variations in gene expression. Is this expected?

Referee #3 (Remarks for Author):

PXR is a significant nuclear receptor and its activation is associated with anti-inflammatory effects. In this paper, the author developed "selective" PXR agonists using a microbial metabolite mimicry strategy. The binding characteristics of the compounds with PXR were evaluated by *in vitro* methods. Two selected compounds, FKK5 and FKK6, were able to transactivate PXR target genes in cell lines, human intestinal organoids and human hepatocytes. The compounds also had protective effects against DSS-induced acute colitis in hPXR humanized mice. The development of the compounds is novel, although the anti-inflammatory activity of PXR agonists is not a surprise based on the literature. The experimental approaches are comprehensive. There are several issues that need to be addressed or clarified.

1. The authors should be careful in using the "selective" term in describing their compounds. "Selective" usually means the biased binding affinity towards one or some member(s) of a family of receptors. Nuclear receptor superfamily consists of many members. According to the limited numbers of nuclear receptors tested in the study, it was an over-statement that FKK5 and FKK6 were "selective" PXR ligands. Their binding capacities/activations on some important nuclear receptors, like FXR, LXR or PPAR γ , were not evaluated in the work. The evaluation of FXR and PPAR γ activation is particularly important because both receptors have been implicated in colitis/colon cancer, and the *in vivo* activation of PXR target genes was not impressive, raising the possibility that the FKK 5 and 6 may have targets other than PXR. For FXR and PPAR γ activation, at least reporter gene assay should be carried out and the expression of their typical target genes should be evaluated *in vivo*.
2. The *in vivo* PXR target gene analysis suggest FKK 5 and 6 are hPXR specific and they do not activate mPXR. The authors should at least use report assay to verify that.
3. Some of the figures need clarification on the meaning of statistical differences. For example, in Fig. 1a, whether the asterisks were from the comparison with the vehicle group or with the lowest concentration of each compound was not indicated. In Fig 1c, no significant differences were observed except for LS180 cells in all three panels. However, changes as large as more than 10000-fold were exhibited. The comparisons should be defined in figure legends. Statistical analysis methods were indicated in Fig. 2 but not in Fig 1a-e.
4. The rationale for choosing the *in vivo* doses should be better explained. The doses should not be decided by compound solubility or DMSO content. Also in WT and humanized mice, the mRNA levels of PXR-target genes were not induced or the induction was not robust, is it possible higher doses might be needed. Besides, the doses given to mice should be expressed in units like mg/kg or g/kg instead of the molar concentrations in the manuscript.
5. Error bars were not presented in the lower panel of Fig. 1d while experiment replicates were claimed.
6. As both FKK6 and rifampicin failed to repress the NF- κ B promoter activity in LS174 cells, the results could not serve as the evidence for FKK6 repression on TLR4-TNF α -NF- κ B axis via PXR binding (Fig. 2e).

7. The authors aimed at developing selective and non-cytotoxic PXR agonists. The selectivity and cytotoxicity were not sufficiently compared between the FKK compounds and the know PXR agonists like rifampicin and SR12813. More discussions should be provided.

1st Revision - authors' response

8th Jan 2020

Please see next page.

Referee #1:

1. Some will find the generic references to 'humanized mice' in the DSS study confusing - it took me a while to spot this meant hPXR mice and not the more common use of the term, human immune system reconstituted mice such as Hu-SCID.

Response: hPXR is now referred to as “mice expressing the human PXR gene”.

2. Figure 5 is quite an important selling point in the paper, but it seems a shame to have done these experiments without characterizing the nature of protection more fully in terms of transcriptional changes and, especially, looking more widely at cytokine changes. For example, any differences in IL-17 and IL-10?

Response: We have now included this data as Fig 5f.

Referee #2:

1. The order of some figures and their panels should be reorganized. Often the result section already described part of figure 2 and 3 then started to talk figure 1d and 1e. This confusion also appeared in other figures.

Response: We have gone through the organization of the figure and text. We detail the flow below to specify that the order of the figures now align with the flow of the text. There are main figures and supplementary figures – each has a sequential position in the text which we list below in the order they appear in the text (Page):

Main Paper

Appendix Figure S1a (Page 4)

Appendix Figure S1b (Page 4)

Appendix Figure S1c (Page 5)

Appendix Figure S1d (Page 5)

Appendix Figure S1e (Page 5)

Fig. 1a (Page 5)

Fig. 1b (Page 5)

Appendix Figure S2a (Page 5)

Appendix Figure S2b (Page 5)

Appendix Figure S2c (Page 5)

Appendix Figure S2d (Page 5)

Appendix Figure S2e (Page 5)

Appendix Figure S2f (Page 6)

Appendix Figure S2g (Page 6)

Appendix Figure S2h (Page 6)

Appendix Figure S2i (Page 6)

Fig. 1c (Page 6-7)

Appendix Figure S3a (Page 7)

Appendix Figure S3b (Page 7)

*Table EV3 (Page 7)**

*Table EV4 (Page 7)**

Appendix Figure S3c and d (Page 7)

Fig. 1d (Page 8)

Appendix Figure S4a & Table EV5 (Page 8)

Fig. 1e (Page 8)

Appendix Figure S4b (Page 9)

Appendix Figure S4c (Page 9)

Fig. 1f (Page 9)

Fig. 2a (Page 9)

Fig. 2b (Page 9)

Fig. 2c (Page 10)

Fig. 2d (Page 10)

Appendix Figure S4d (Page 10)

Appendix Figure S5a (Page 10)

Appendix Figure S5b (Page 10)

Appendix Figure S5c (Page 10)

Fig. 2e (Page 10)
Fig. 3a (Page 11)
Fig. 3b (Page 11)
Fig. 3c (Page 11)
Fig. 3d (Page 12)
Fig. 3e (Page 12)
Fig. 3f (Page 12)
Appendix Figure S6a (Page 11)
Fig. 4a (Page 12)
Fig. 4b (Page 12)
Fig. 4c (Page 12)
Fig. 4d: 36 h; Appendix Figure S6b: 60 h (Page 13)
Fig. 4e: 36 h; Appendix Figure S6c: 60 h (Page 13)
Fig. 4f (Page 13)
Appendix Figure S6d (Page 14)
Fig. 5a, b
Fig. 5c
Fig. 5d, e
Fig. 5f
Fig. 6a–e

Experimental Section (Online Methods)

Appendix Figure S5a & b (Page 19)

Supplementary Methods

Figure EV1: Overview of synthetic routes

Figure EV2: NMR traces of FKK1–10

Table EV1 and Table EV2 (Page 72)

* *Table EV1 and Table EV2 are referred to in the supplementary document (results)*

Note: Back references of figures are part of discussion and comparisons made.

2. Figure 1d bottom, the quantitative PCR data should be provided with mean values and standard deviations.

Response: The mean values are shown now in Fig. 1d. The SD values are not shown as they will crowd the figure, but the SD is shown as error bars.

3. Figure 3a and 3d, it's unclear what the * indicates for; which groups were compared. It would be helpful if arrows or other indicators were placed in fig 3c and 3f to clarify the localization of NF- κ B.

Response: In the legend for Fig. 3 we have inserted “ * for cytokines versus no cytokines, compares either the 12 hr, 6hr, 4hr or 2 hr equivalent time-point(s) as indicated; * for cytokines versus cytokines + FKK compound, compares either the 12hr, 6hr, 4hr or 2hr equivalent time-point(s) as indicated. **c**, Arrows represent localization of NF κ B. “

4. Figure 4d showed very large variations in gene expression. Is this expected?

Response: The large variation of the values comes from one of the three mouse tissues with values ~3-times that of the other two. The error bar is a 95%CI limit around the mean which exaggerates the deviation considerably (see the same figure below plotted as mean and SD on the left panel and mean and 95%CI right panel):

We expect and have shown the true values likely lie within the 95% CI bound and hence there isn't much difference between the groups. We have used 95%CI as a means to show variation in **ALL** the figures pertinent to mouse studies. The exception is Fig S6c which will have the lower confidence bound outside the graph and so for visual appeal is expressed as mean (SD).

Referee #3:

1. The authors should be careful in using the "selective" term in describing their compounds. "Selective" usually means the biased binding affinity towards one or some member(s) of a family of receptors. Nuclear receptor superfamily consists of many members. According to the limited numbers of nuclear receptors tested in the study, it was an over-statement that FKK5 and FKK6 were "selective" PXR ligands. Their binding capacities/activations on some important nuclear receptors, like FXR, LXR or PPAR γ , were not evaluated in the work. The evaluation of FXR and PPAR γ activation is particularly important because both receptors have

been implicated in colitis/colon cancer, and the in vivo activation of PXR target genes was not impressive, raising the possibility that the FKK 5 and 6 may have targets other than PXR. For FXR and PPAR γ activation, at least reporter gene assay should be carried out and the expression of their typical target genes should be evaluated in vivo.

Response: We agree. The word “selective” has been deleted to describe the agonists (abstract line 6). On Page 7, the word “PXR-selective” refers to a comparison of the FKK compounds, in that, FKK6 is more PXR selective than say FKK2 or 9 (which activate PXR and AhR) (Page 7). Thus, in those references the word “selective” is retained. The word “PXR-specific” is retained because it refers to the relative specificity of FKK compounds, in particular FKK5 and FKK6, on PXR activity versus other nuclear receptors tested (see abstract, page 4, page 6).

We have now evaluated the effects of FKK5 and FKK6 on FXR (TR-FRET assay: Supplementary Fig. 2g) and PPAR γ (Cell line reporter assay: Supplementary Fig. 2h) activation. Neither FKK5 or FKK6, have any receptor activation potential. It is to be noted that the in vivo activation of FKK6 in mice was performed using an “ad hoc” dose concentration and schedule and was simply done to show proof-of-concept. The drug dose and schedule were not optimized for in vivo delivery and receptor activation.

2. The in vivo PXR target gene analysis suggest FKK 5 and 6 are hPXR specific and they do not activate mPXR. The authors should at least use report assay to verify that.

Response: We agree. We have now included mPXR reporter assay (Supplementary Fig. 2i).

3. Some of the figures need clarification on the meaning of statistical differences. For example, in Fig. 1a, whether the asterisks were from the comparison with the vehicle group or with the lowest concentration of each compound was not indicated. In Fig 1c, no significant differences were observed except for LS180 cells in all three panels. However, changes as large as more than 10000-fold were exhibited. The comparisons should be defined in figure legends. Statistical analysis methods were indicated in Fig. 2 but not in Fig 1a-e.

*Response: We have now clarified this in the legend as all comparisons are made to control (vehicle exposed) (Fig. 1a) or mock-transfected or knockout cells as relevant to Fig 1c. Fig 1a statistics is * $p < 0.05$, one-way ANOVA with Dunnett's post hoc test. *significant over vehicle (DMSO) control. Fig 1b statistics is * $p < 0.05$, one-way ANOVA with Dunnett's post hoc test. *significant over vehicle (DMSO) control. Fig 1c statistics is *, # $p < 0.05$, two-way ANOVA with Tukey's post hoc test. *significant over vehicle control. #significant over the same treatment in corresponding mock transfected or knock-out cells. Fig. 1 d-f do not have statistics.*

4. The rationale for choosing the in vivo doses should be better explained. The doses should not be decided by compound solubility or DMSO content. Also, in WT and humanized mice, the mRNA levels of PXR-target genes were not induced, or the induction was not robust, is it

possible higher doses might be needed. Besides, the doses given to mice should be expressed in units like mg/kg or g/kg instead of the molar concentrations in the manuscript.

*Response: We have provided the rationale for choosing in vivo doses in the revised manuscript and this rationale was inadvertently missing in our first submission. Since we do not know the pharmacokinetics of FKK6 in vivo, we estimated based on the maximum bioavailability indole compounds could have in vivo (~ 30-90%), that at least a feces concentration of 50 μ M will be found at the end of 3-5 doses in vivo (this is greater than the 10 μ M used in cell culture experiments). For the colitis study, the rationale for choosing this dose (0.4 mg/kg/d for 10 days) is based on the extrapolation of total dose delivered over 10 days to mice (~ 4 mg/kg), which is equivalent to that delivered to mice for determining target gene expression over 3-5 doses (3-5 mg/kg). The sample size of $n = 3$ was chosen arbitrarily as means to explore if at least one mouse responded to FKK6. All doses are now expressed in mg/kg in addition to molar concentrations. In hPXR mice, FKK6 induces cyp3a11 mRNA ~ 4-fold over vehicle. We do not expect induction of mRNA in wild-type mice (< 2-fold, we also show that mPXR is not activated by FKK6, see Supplementary Fig. 2i). In hPXR mice, in the intestine, rifampicin is barely able to induce CYP3A4 (homolog of cyp3a11)₁ (see Fig 2 of reference). Indeed, in another mouse model expressing the human PXR transgene, the mean colon induction of cyp3a11 mRNA is 2.5 fold with rifaximin (see Fig 6 of paper)₂; however, up to 4-fold can be observed **in small intestines BUT not large intestines** depending on the hPXR model used₃. Thus, in our model FKK6 actually does get to the colon and can induce PXR target, cyp3a11 ~ 4-fold over vehicle suggesting a relatively robust gene expression. While we do not disagree with the possibility that higher doses are needed, we are currently performing independent studies to look at dose-gene response relationships with respect to PK-PD analysis. We feel that at the present time, this data is beyond the scope of our initial proof-of-concept.*

5. Error bars were not presented in the lower panel of Fig. 1d while experiment replicates were claimed.

Response: The error bars (as SD) are now included and mean fold values expressed within the figure panel.

6. As both FKK6 and rifampicin failed to repress the NF- κ B promoter activity in LS174 cells, the results could not serve as the evidence for FKK6 repression on TLR4-TNF α -NF- κ B axis via PXR binding (Fig. 2e).

*Response: Yes, we agree with our findings as these cells were **LS174T cells with a knockout of the PXR receptor** (see Fig 1e legend). Thus, there is no response of FKK6 on the repression of NF- κ B promoter activity which supports “the evidence for FKK6 repression on TLR4-TNF α -NF- κ B axis via PXR binding”.*

7. The authors aimed at developing selective and non-cytotoxic PXR agonists. The selectivity and cytotoxicity were not sufficiently compared between the FKK compounds and the known PXR agonists like rifampicin and SR12813. More discussions should be provided.

Response: We have expanded the discussion to explain why our FKK analogs are more selective and non-toxic than known PXR prototypical ligands, rifampicin, rifaximin and SR12813.

References

- 1 Kobayashi, K. *et al.* CYP3A4 Induction in the Liver and Intestine of Pregnane X Receptor/CYP3A-Humanized Mice: Approaches by Mass Spectrometry Imaging and Portal Blood Analysis. *Molecular pharmacology* **96**, 600-608, doi:10.1124/mol.119.117333 (2019).
- 2 Cheng, J. *et al.* Therapeutic role of rifaximin in inflammatory bowel disease: clinical implication of human pregnane X receptor activation. *The Journal of pharmacology and experimental therapeutics* **335**, 32-41, doi:10.1124/jpet.110.170225 (2010).
- 3 Ma, X. *et al.* Rifaximin is a gut-specific human pregnane X receptor activator. *The Journal of pharmacology and experimental therapeutics* **322**, 391-398, doi:10.1124/jpet.107.121913 (2007).

3rd Editorial Decision

30th Jan 2020

Thank you for the submission of your revised manuscript to EMBO Molecular Medicine, and my apologies for the delay in getting back to you following the holiday season. We have now received the enclosed reports from the three referees who had originally reviewed your manuscript. As you will see, they are all supportive of publication, and I am thus pleased to inform you that we will be able to accept your manuscript pending the following final editorial amendments.

***** Reviewer's comments *****

Referee #1 (Comments on Novelty/Model System for Author):

I was enthusiastic about the earlier draft of this manuscript, though with some reservations about limitations. This version is much strengthened

Referee #1 (Remarks for Author):

This resubmission is much strengthened

Referee #2 (Remarks for Author):

The authors have adequately addressed the reviewers' comments.
Accept

Referee #3 (Remarks for Author):

The authors have adequately addressed my comments and as a result, the manuscript has improved from its previous version.

2nd Revision - authors' response

4th Feb 2020

The authors performed the requested editorial changes.

YOU MUST COMPLETE ALL CELLS WITH A PINK BACKGROUND ↓
PLEASE NOTE THAT THIS CHECKLIST WILL BE PUBLISHED ALONGSIDE YOUR PAPER

Corresponding Author Name: Sridhar Mani
Journal Submitted to: EMBO Molecular Medicine
Manuscript Number: EMM-2019-11621-T